# A European-Chinese Exploration: Part 2—Urban Ecosystem Service Patterns, Processes, and Contributions to Environmental Equity under Different Scenarios

Wanben Wu [1,2], Xiangyu Luo [3], Julius Knopp [2], Laurence Jones [4] and Ellen Banzhaf [2,*]

1 Ministry of Education Key Laboratory for Biodiversity Science and Ecological Engineering, National Observations and Research Station for Wetland Ecosystems of the Yangtze Estuary, Shanghai Institute of EcoChongming (SIEC), Fudan University, Shanghai 200433, China; 19110700121@fudan.edu.cn
2 Department of Urban and Environmental Sociology, UFZ—Helmholtz Centre for Environmental Research, Permoserstr. 15, 04318 Leipzig, Germany; julius.knopp@ufz.de
3 Ministry of Education Key Laboratory for Earth System Modelling, Department of Earth System Science, Tsinghua University, Beijing 100084, China; luoxy17@mails.tsinghua.edu.cn
4 UK Centre for Ecology & Hydrology, Environment Centre Wales, Deiniol Road, Bangor LL57 2UW, UK; lj@ceh.ac.uk
* Correspondence: ellen.banzhaf@ufz.de

**Abstract:** Urban expansion and ecological restoration policies can simultaneously affect land-cover changes and further affect ecosystem services (ES). However, it is unclear whether and to what extent the distribution and equity of urban ES are influenced by the stage of urban development and government policies. This study aims to assess the quantity and equity of ES under different scenarios in cites of China and Europe. Firstly, we used the Conversion of Land Use and its Effects at Small regional extent (CLUE-S) model to simulate future land cover under three scenarios: business-as-usual (BAU), a market-liberal scenario (MLS), and an ecological protection scenario (EPS). Then using ecosystem service model approaches and the landscape analysis, the dynamics of green infrastructure (GI) fraction and connectivity, carbon sequestration, and $PM_{2.5}$ removal were further evaluated. The results show that: (1) over the past 20 years, Chinese cities have experienced dramatic changes in land cover and ES relative to European cities. (2) Two metropolises in China, Shanghai and Beijing have experienced an increase in the fraction and connectivity of GI and ES in the long-term built-up areas between 2010 and 2020. (3) EPS scenarios are not only effective in increasing the quantity of ES but also in improving the equity of ES distribution. The proposed framework as well as the results may provide important guidance for future urban planning and sustainable city development.

**Keywords:** CLUE-S; scenario analysis; equity; green infrastructure fraction; carbon stock; $PM_{2.5}$ removal

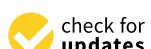



## 1. Introduction

It is widely accepted that ecosystems provide multiple benefits for human well-being via ecosystem services (ES). The higher the population density, the more important are the supplies of services, and this is especially the case in urbanized regions. ES provided by the hinterlands surrounding cities, and within cities by green and blue spaces, both have benefits for the urban population [1]. However, rapid urbanization and population growth can easily lead to the exploitation and degradation of ecosystems [2], witnessed by findings of the Millennium Ecosystem Assessment [3,4], which showed most ES having been in decline over the past years due to human activities [5]. Specifically, urbanization processes continually shape the quantity and quality of urban blue-green space, determine the location of nature reserves, or, alternatively, influence people's desires and needs for ES [6]. Meanwhile, economic growth drivers and policy decisions can also influence management and decision making to change the distribution patterns of ES. The impacts of

policies on ES are even more multifaceted and complex. First, policies directly influence the direction of urban development and the balance between economic and ecological priorities. Additionally, regions at different stages of development have different needs and therefore set different priorities for planning. In this context, the mapping and modelling of ES have become an important way to help scientists, managers, and policymakers better understand and manage urban ecological resources, which will further contribute to the restoration of urban ecosystems and the achievement of sustainable development goals (SDGs), especially SDG 11 [7,8].

Interestingly, the way in which policies affect urban ES differ substantially between China and Europe. China has experienced rapid urbanization over the past four decades, with its urban population increasing from 170 million in 1978 to 837 million in 2018, and its urbanization level (the proportion of the population living in urban areas) has increased from 18% to 60% during this period [9]. According to the United Nations Development Programme (UNDP), China's urbanization level will reach 70% in 2030 [10,11]. In addition to rapid urbanization, China has launched a myriad of sustainability initiatives to promote the transition towards urban sustainable development [12]. For example, the Beijing Plains Afforestation Program has done a great job of improving air quality, mitigating the urban heat island effect, and preventing soil erosion [13]. Urban green space is also increasing, with 65% of the 117 medium and large cities in China showing increased greening in their urban centers between 2010 and 2019 [14]. All of the above studies demonstrate the efforts and efficiency of Chinese policies to restore urban ES.

In Europe, the industrial revolution was the major driver for urbanization processes, and urban agglomerations started as early as the 18th century and have mostly reached a saturation stage. For this reason, Europe has been an urban-centered continent for centuries. The urbanization level in Europe has risen at a much slower pace than in China during the last decades, with World Bank statistics showing a rise from 70.8% in 2000 to 75% in 2020 [15]. Across European regions, different historical and political contexts have caused a high degree of heterogeneity in urbanization patterns, with a diversity of small and medium-sized cities with low growth patterns and only very few megacities [16]. European cities have undergone a series of low-density discontinuous developments since the 1950s [17]. Europe has also had active policies to limit urban sprawl, for example by creating "green belts" around cities to protect urban growth or by defining a thirty-hectare target for sealing surfaces to prevent extreme construction activity and promote urban densification. Despite these policies, urban sprawl continues, and urbanization processes intensify [18]. Different urban development patterns affect the distribution and dynamics of ES at multiple scales. Therefore, comparing the dynamics of Chinese and European urban ES can better reflect the impacts of different urbanization patterns and stages as well as policy instruments on the development of ES, thus improving our understanding of the coupled social–ecological system relationship. Unfortunately, such comparative studies are sorely lacking.

Scenario analysis can provide a more meaningful theoretical basis and decision reference for balancing economic development and ecological conservation and, therefore, has received increasing attention in urban research [19]. For example, Liu, et al. [20] constructed several scenarios covering policy and climate change, including the one-child policy and carbon tax policy, and projected the land use distribution under various scenarios, which evaluated impacts on carbon sequestration, soil conservation, and water yields. Based on the aesthetic value and the recreation value of nature reserves, Qin, et al. [21] combine social and natural factors from the perspective of ES and select priority protected areas by comparing conservation efficiency under multiple scenarios. Gao, et al. [22] used a CA-Markov model to analyze the land use and ecosystem service values of Shijiazhuang, China in 2030 under a natural development scenario, farmland protection scenario, and an ecological protection scenario. These studies provide an important reference for future urban land cover and ecosystem service estimation.

In this paper, our central goal is to understand future urbanization patterns and their effects on ES quantity and equity under a range of policy scenarios. To achieve this goal, the decisive steps towards this are as follows: (1) Predict ES distribution patterns over the next decade under three scenarios. (2) Analyze the differences and characteristics of ES dynamics provided by different phases of development. (3) Explore changes in environmental equity of ES distribution due to urbanization. (4) Assess how land-cover dynamics have led to differences in green infrastructure (GI) and ES changes.

## 2. Dataset and Materials

### 2.1. Study Area

Our analysis comprises three Chinese cities (Beijing, Shanghai, and Ningbo) as well as three European Cities (Paris region—France, Aarhus—Denmark, and Velika Gorica—Croatia) (Figure 1). Among the three Chinese cities, Beijing is the capital of China, with an area of 16,410 km$^2$ and a population of 21.7 mio. Between 2000 and 2020, Beijing experienced significant urbanization with an increase in the built-up area from 1640 km$^2$ to 2859 km$^2$. Much larger in terms of population is Shanghai, with a total area of 6340 km$^2$ and a population of 24.2 mio. It has also experienced significant urban expansion over the past 20 years, with the built-up area increasing from 1414 km$^2$ to 2793 km$^2$ by 2020. While Ningbo is smaller than Beijing and Shanghai in population (5.7 mio.), it is also rapidly gaining built-up area, growing from 1023 km$^2$ in 2000 to 1857 km$^2$ in 2020. These cities represent typical Chinese urbanization patterns of megacities and cities with over a million inhabitants. Amongst the European study sites, the Paris region is the biggest urban agglomeration, and one of the few megacities in this continent. It is France's capital and home to 18.2% of the country's population (12 mio.) in which the built-up area grew from 1680.4 km$^2$ to 1884.7 km$^2$ between 2000 and 2020. With 273,000 inhabitants, Aarhus, Denmark, has experienced slow urbanization over the past 20 years, changing its built-up area from 77 km$^2$ to 85 km$^2$. In Croatia, Velika Gorica has 64,000 residents, with an average population density of 272 people per km$^2$, and its built-up area has been decreasing from 23 km$^2$ to 17 km$^2$ during the past twenty years [23]. So, the European study sites cover the types of a megacity, a typical mid-sized city, and a town.

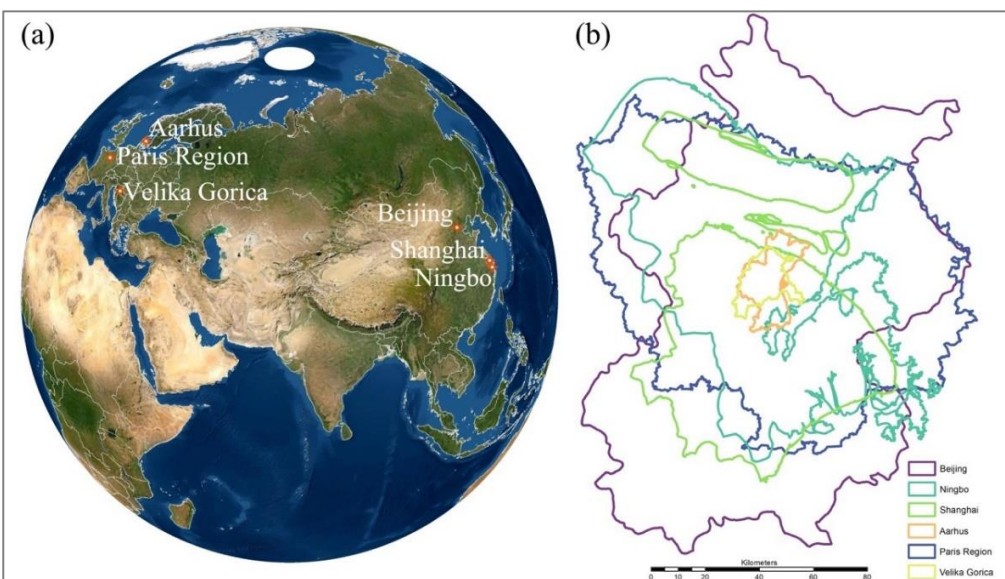

**Figure 1.** (**a**) Location of the six study cities and (**b**) comparison of their urban forms.

### 2.2. Data Source and Processing

#### 2.2.1. Refined Land-Cover Maps

Europe and China Refined Land Cover (ECRLC) is a 30-m Landsat-based land-cover database spanning 3 decades (2000, 2010, and 2020) for Aarhus, Paris region, and Velika Gorica in Europe and Beijing, Shanghai, and Ningbo in China. The overall classification accuracies range between 73% and 95% for different points in time and cities [23]. These ECRLC products were utilized for future land-cover simulation in 2030 (see Section 3.1) and also as the land cover input for GI and ES evaluation.

#### 2.2.2. Driving Factors of Land-Cover Spatial Distribution

In order to simulate land cover in the future, several driving factors including natural geographical, location, and socio-economic factors were considered spatial distribution drivers. Table 1 shows the year of acquisition of the data used, the spatial resolution, and the data source references. Among them, the vector data of primary road, second class road, and river were obtained from OpenStreetMap [24]. The Euclidean distance algorithm in ArcGIS (Version 10.8) was used to calculate the distance of each pixel from the primary roads, the second level roads, and the rivers. The night light intensity was available from the Defense Meteorological Program (DMSP) and was also normalized based on the annual maximum normalized difference vegetation index (NDVI) to eliminate the oversaturation phenomenon [14]. Finally, all data were resampled to 60 m using the nearest neighbor algorithm in ArcGIS (Version 10.8) for further land cover simulation.

**Table 1.** Variable system for evaluation land suitability.

| Variables | Year | Resolution | Reference |
|---|---|---|---|
| Altitude (m) | 2010 | 30 m | [25] |
| Slope (°) | 2010 | | |
| Population | 2010 | 100 m | [26] |
| Population growth (%) | 2010–2020 | | |
| Distance to primary road (km) | 2010 | | |
| Distance to second level road (km) | 2010 | vector data | [24] |
| Distance to river (km) | 2010 | | |
| Night light intensity | 2010 | 300 m | [27] |

#### 2.2.3. Nature Reserve Area

Nature reserves were treated as protected areas, and the land cover of the area did not change when simulating future land cover. In this study, nature reserves in Europe were tracked using the World Database on Protected Areas (WDPA) [28], while nature reserves for the three Chinese cities were derived from the Specimen Resource Sharing Platform of China Nature Reserve [29]. All nature reserve data were vector data, and when used as input data for the CLUE-S model, we converted the vector data to raster data with a spatial resolution of 60 m for land cover simulation.

### 3. Method

In this study, a comprehensive framework was constructed to evaluate GI and ES in the past and future. The procedure is shown in Figure 2 and consists of three main parts, including (i) the simulation of land cover for 2030 under three different development scenarios using the CLUE-S model; (ii) evaluating the ES based on several ES evaluation models; and (iii) evaluating the spatio-temporal dynamics and distribution equity of ES.

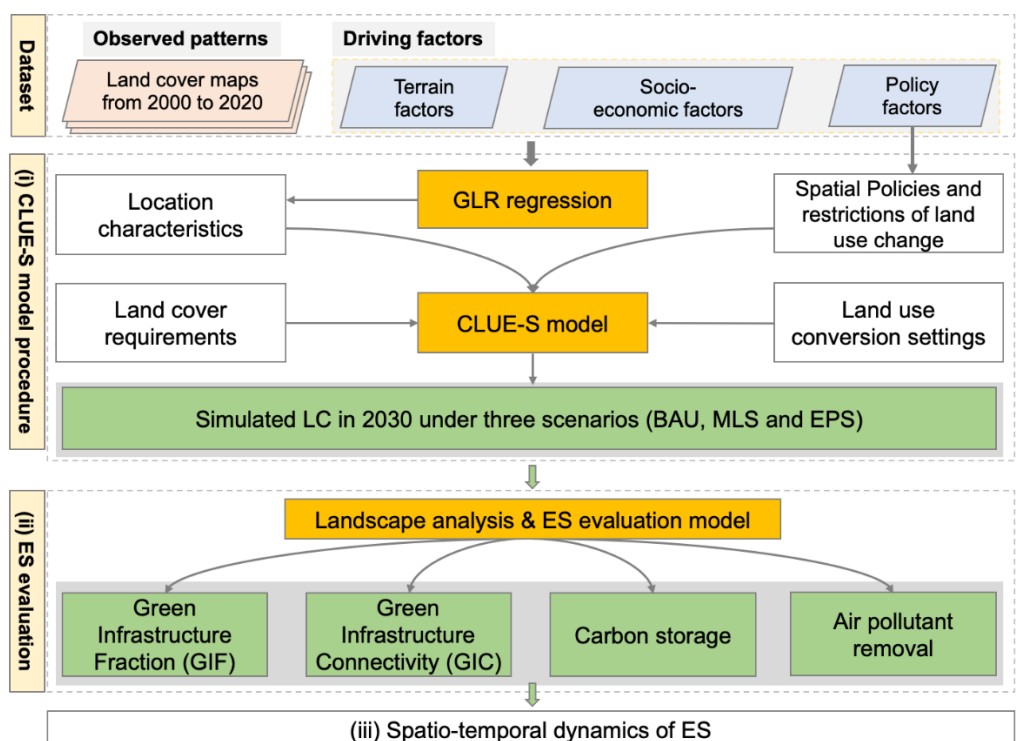

**Figure 2.** Framework from past to future for ES evaluation (BAU = business-as-usual scenario; MLS = market-liberal scenario; EPS = ecosystem protection scenario).

### 3.1. Future Land Cover Simulation

#### 3.1.1. CLUE-S Model

The CLUE-S model has been widely used for future land cover/use simulation in previous research [30–32]. It is a dynamic, spatially explicit LULC model, which can consider location characteristics of land cover distribution; the demand of requirements, spatial policies, and restrictions of land-cover changes; and also land cover conversion settings. In addition, various scenarios could also be considered during the simulation. In this study, the spatial resolution of $60 \times 60$ m$^2$ was used for the simulation. It was carried out based on the R package "lulcc" [33]. The parameter settings for each part of the simulation are described below.

For the location characteristics, random forest machine learning methods were used to quantify the relationships between land-cover pattern and explanatory factors, including terrain and socioeconomic factors (Table 1), to determine the probabilities of the distribution of each land cover type [30]. All of the factors were normalized to the range 0 to 1 by minimum–maximum linear transformation. The relative operational characteristics (ROC) reflects the goodness of fit, with ROC values ranging from 0.5 to 1. When the ROC is greater than 0.7, the goodness of fit is acceptable [34].

Spatial policies and restricted zones were considered as regions where it cannot be changed [35]. In this research, this includes nature reserves. Finally, the land cover conversion was calculated based on the transition matrix of land cover during 2010 to 2020; the conversion matrix means the conversion direction between land cover types, which ranges from 0 (prohibit conversion) to 1 (allowed conversion).

#### 3.1.2. Simulation Scenarios Setting

To estimate different development patterns for the year 2030, three different scenarios were created. The first is the so-called business-as-usual (BAU) that predicts land-cover change progressing in a linear way following the current trend without new policies influencing changes neither towards environmental protection nor towards economic growth and built expansions. The second is market-driven by economic prosperity (MLS)

in which urbanization continues to expand, at the cost of environmental protection. The third is set towards environmental protection (EPS), which assumes a less extreme growth rate with the target of a gain or at least maintaining of ES for urban dwellers.

In this study, the land cover demands were calculated based on three different scenarios. Specifically, under the BAU, the demand areas of land cover were consistent with the land-cover change trend based on land cover maps from 2010 to 2020. Under the MLS, the change trend of built-up area was increased by 50% compared with BAU [36]. Finally, under EPS, the demand of ecological lands was increased by 50% when it was increasing during 2010 to 2020; otherwise, the decrease rate was slowed down by 50%.

### 3.1.3. Land-Cover Simulation Accuracy Evaluation

To evaluate the rationality and accuracy of the land cover simulation framework, we validated the allocation model in two steps; firstly, we could obtain the accuracy of each GLR model, and secondly, we could verify the rationality of the model by comparing the simulation results of change over the period 2010 to 2020, with observations of land cover in 2020. The receiver operator characteristic (ROC) measures the degree of fitting of the GLR model. This index can be used to assess the accuracy of the model. Specifically, ROC values vary from 0.5 to 1, where 0.5 indicates a completely random model and 1.0 indicates a perfect fit [37]. GLR models with ROC values above 0.7 are considered good [38,39]. Based on land cover data in 2010, we simulated land cover distribution in 2020 for all of the six cities and further compared them with the observations; then the overall accuracy (OA) and Kappa indices [40] were used to quantify the accuracy between the simulation and the observation maps. The overall accuracy and Kappa coefficients are calculated as follows:

$$Overall\ accuracy = \frac{Number\ of\ correct\ pixels}{Total\ number\ of\ pixels} \times 100 \tag{1}$$

$$Kappa = \frac{N \sum_{i=1}^{r} x_{ii} - \sum_{i=1}^{r} (x_{i+} \times x_{+i})}{N^2 - \sum_{i=1}^{r} (x_{ii} \times x_{+i})} \tag{2}$$

where $N$ represents the number of validation samples, $x_{ii}$ represents the number of samples in row $i$ and column $i$ in the confusion matrix, $x_{i+}$ represents the sum of all samples in row $i$, and $x_{+i}$ represents the sum of all samples in column $i$ in the confusion matrix.

### 3.2. Spatially Explicit Indicators for GI and ES
### 3.2.1. GI Fraction

In this study, the GI fraction (GIF) is used as an important indicator of ES in a 1-km × 1-km unit during urban development. In this study, GI comprises green spaces and cropland because both categories support ES significantly. The specific calculation method is shown as below

$$GIF = \frac{A_{GI}}{A_{total}} \tag{3}$$

where $A_{GI}$ represents the *GI* area, and $A_{total}$ represents the area of each unit, which is set as 1 km$^2$ in this study.

### 3.2.2. GI Connectivity

Our study examines GI connectivity (GIC) within a 1 km × 1 km unit based on land cover observations and simulations. The patch cohesion index was calculated to evaluate natural land cover connectivity using package "landscapemetric" in R version 4.0.2 [41]. The index takes values between 0 and 100, with larger GIC values indicating better landscape connectivity.

$$GIC = 1 - \left( \frac{\sum_{j=1}^{n} p_{ij}}{\sum_{j=1}^{n} p_{ij} \sqrt{a_{ij}}} \right) \cdot \left( 1 - \frac{1}{\sqrt{Z}} \right)^{-1} \cdot 100 \tag{4}$$

where $p_{ij}$ and $a_{ij}$ are the perimeter and area of each patch, respectively, and $Z$ is the number of *GI* pixels.

### 3.2.3. Carbon Stock

To quantify the amount of carbon stock in the six cities, we estimated it using carbon density in different land use types and their area through the carbon stock module of the Integrated Valuation of Environmental Services and Tradeoffs (InVEST) model [42]. In this module, carbon stock contains four carbon pools, including aboveground carbon pool, belowground carbon pool, soil organic carbon pool, and dead matter organic carbon pool.

$$CS = \Sigma CD_i \cdot A_i \tag{5}$$

where $CS$ is the total amount of carbon stock in a year (t), $\Sigma CD_i$ is the total carbon density of four carbon pools in land use type $i$ (t/ha), and $A_i$ is the acreage of land use type $i$ (ha).

Considering the differences of vegetation composition and structure and management mode in urban green space between Chinese cities and European cities, the appropriate parameters were selected. The carbon densities of the four carbon pools in various land cover types of Chinese cities and European cities were derived from the previous literature by considering the characteristics of our land cover categories [43–46]. The carbon densities used in this study are detailed in Table A1.

### 3.2.4. Air Pollutant Removal

To quantify the amount of PM$_{2.5}$ removal by urban vegetation, we gathered the data for PM$_{2.5}$ concentration derived from the Global Estimates of Fine Particulate Matter [47] and then applied the method developed by [48,49] to quantify the PM$_{2.5}$ removal for the six cities. In addition, to simulate the air pollutant removal service in the future under different scenarios, we took the average PM$_{2.5}$ concentration of the latest five years to calculate the removal rate.

$$PM_r = \alpha \cdot PM_c + \beta \tag{6}$$

where $PMr$ is the quantity of PM$_{2.5}$ removed per unit area of forest and grassland per year (kg ha$^{-1}$ yr$^{-1}$), $PMc$ is the annual concentration of PM$_{2.5}$ (µg m$^{-3}$), and $\alpha$ and $\beta$ are the regression coefficients, where their values are 1.1664 and 0.4837 ([48,49]), respectively.

$$TR = PM_r \cdot A \tag{7}$$

where $TR$ is the total amount of PM$_{2.5}$ removed by woodland in a year (kg), and $A$ is the area of forest and grassland (ha).

### 3.3. Spatial Distribution Characteristics of GI and ES

3.3.1. Urban Development Phases Detection

For a more effective distinction between urban development and policy-driven ecological restoration, and to examine how land cover and ES differ across different urban development gradients, we divided each city into three sections: long-term built-up areas (built-up areas since 2000), new built-up areas (built-up areas from 2000 to 2020), and non-built-up areas. According to the land cover observations discussed in Section 2.2.1, built-up areas for 2000 and 2020 were extracted. Additionally, we used morphological and kernel density estimation methods to fill in the internal gaps between the long-term built-up area and the newly built-up area. Figure A1 shows the three phases in each city.

3.3.2. Equity Measurement of ES

The GINI index was used as an indicator to measure the degree of equity in the spatial distribution of ES, which can be calculated based on the Lorenz curve [50] as follows:

$$\text{Gini} = A/(A + B) \tag{8}$$

where A is the area between the line of equal distribution and B represents the area under the Lorenz curve. The GINI coefficient takes values between 0 and 1. A larger Gini coefficient value indicates greater inequity, i.e., a Gini coefficient of 0 indicates absolute equity, while absolute inequity is represented when the Gini coefficient is equal to 1. In this study, we used the "ineq" package [51] in R to calculate the Gini coefficient.

### 3.4. Sensitivity as the Synthesis of ES Dynamics

We calculated a sensitivity index (SI) [52] to assess the impact of land-cover change on ES changes. In particular, land-cover changes had a positive impact on ES changes when SI > 0; otherwise, they exerted a negative impact. SI is calculated with the following formula.

$$CDI = \frac{\sum_{i=1}^{4} \Delta LC_i}{\sum_{i=1}^{4} LC_i} \times \frac{1}{T} \tag{9}$$

$$SI = \frac{(ES_{end} - ES_{start})/ES_{start}}{CDI} \tag{10}$$

where *CDI* is the degree of land-cover change within a given period and represents the area of land cover *i* that has changed, *LCi* is the area of land cover *i*, and *T* represents the range of years. The *ES_start* and *ES_end* indicators correspond to the *ES* in the start year and end year during the study period, respectively.

## 4. Results

### 4.1. Land Cover Simulation Result

As shown in Figures 3 and 4, there is a significant difference between the land cover patterns in 2020 and the three simulated scenarios in 2030. Figure A2 shows the land cover conversion for different scenarios from 2020 to 2030.

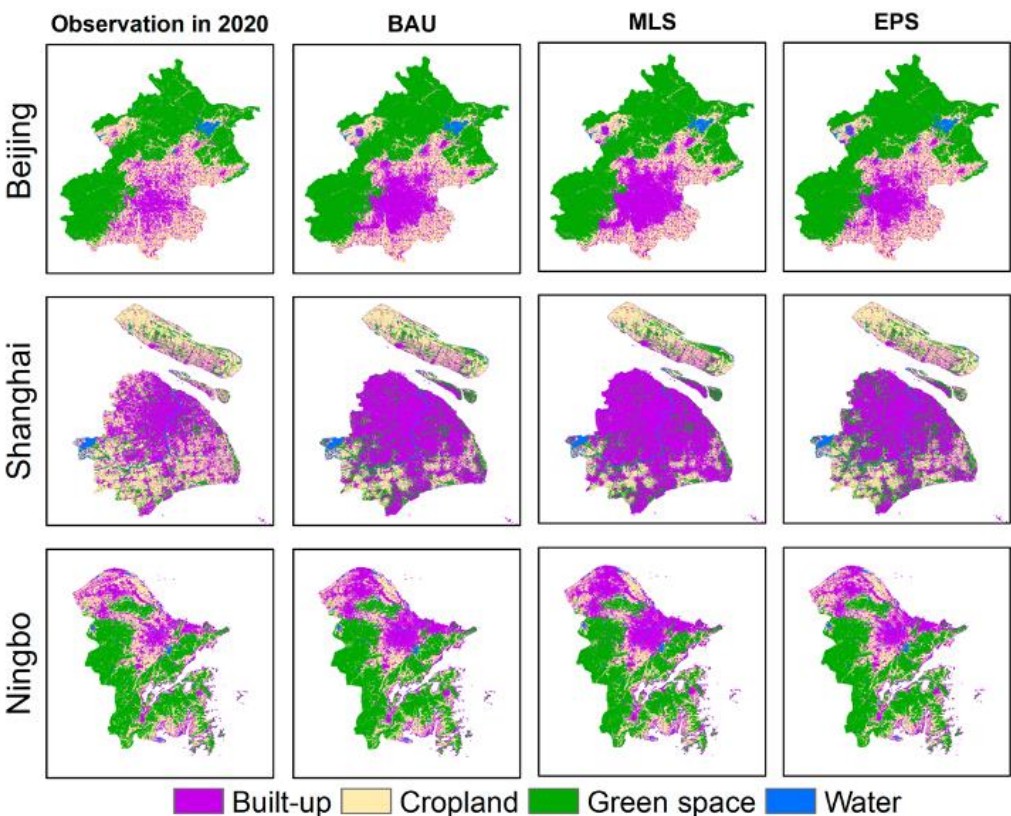

**Figure 3.** Land-cover observation results for 2020 and simulated land-cover maps for 2030 under three scenarios for three cities in China.

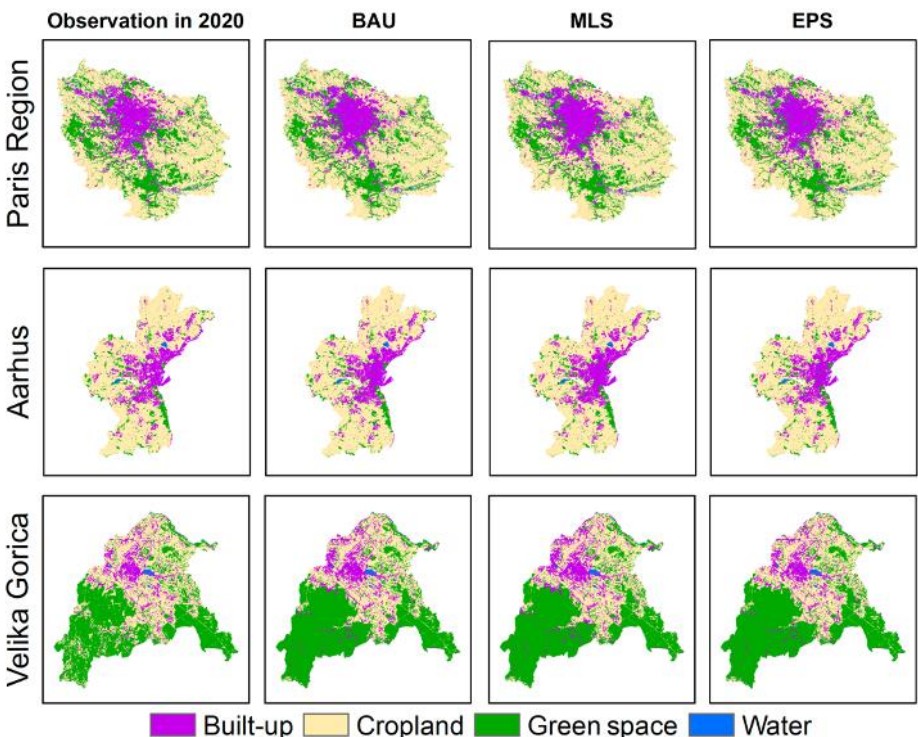

**Figure 4.** Land-cover observation results for 2020 and simulated land-cover maps for 2030 under three scenarios for three cities in Europe.

For the three Chinese cities, the built-up area shows rapid growth in all three development scenarios. Specifically, in Beijing, the built-up area grew up to the 643.2 km$^2$ (BAU), 965 km$^2$ (MLS), and 322 km$^2$ (EPS) scenarios by 2030, where almost all (98.7%, 92.3%, and 99.3%) new built-up space was transformed from arable land. In Shanghai, the built-up area increased to 819 km$^2$, 1223 km$^2$, and 413 km$^2$ under the BAU, MLS, and EPS scenarios by 2030. It is worth noting that in all three scenarios in Shanghai, the area of forest grows, mainly converted from cropland, with forest area increasing by 371 km$^2$, 224 km$^2$, and 518.5 km$^2$ in BAU, MLS, and EPS, respectively. In Ningbo, the built-up area increased by 476 km$^2$ (BAU), 720 km$^2$ (MLS), and 215 km$^2$ (EPS). Under the BAU and MLS scenarios, 151 km$^2$ and 189 km$^2$ of forest were converted to cropland, respectively. Overall, the three cities will continue to experience extensive urban expansion in terms of built-up area over the next ten years, with average growth rates of 23.8% (BAU), 35.9% (MLS), and 11.6% (EPS). In the process, cropland will become less available, with average reductions of 24.1% (BAU), 28.6% (MLS), and 19.5% (EPS) in the three cities. The green space area in the three cities shows different trends from 2020 to 2030, with Beijing and Ningbo showing relatively few changes in the green space area under the three scenarios, while in Shanghai, the green space area increases to varying degrees under the three scenarios, at 39.5 km$^2$ (BAU), 19.9 km$^2$ (MLS), and 58.8 km$^2$ (EPS).

The land-cover patterns of the three European cities for the year 2030 are shown in Figure 4. The Paris region and Aarhus exhibited relatively little urban expansion between 2020 and 2030, with Paris showing built-up area growths of 111 km$^2$ (BAU) with a growth rate of 5.5%, 116 km$^2$ (MLS) with a growth rate of 8.3%, and 56 km$^2$ (EPS) with a growth rate of 2.8%. Velika Gorica demonstrated a relatively significant urbanization intensity, with built-up area growths of 6.6 km$^2$ (19.6%) (BAU), 10.1 km$^2$ (29.9%) (MLS), and 3.1 km$^2$ (9.3%) (EPS), respectively. The average growth rates of built-up area in the three municipalities range from 10% (BAU) to 15.2% (MLS) and then drop down to 5% (EPS). The average decreases in cropland area are 6.8% (BAU), 7.6% (MLS), and 6.1% (EPS). There are differences in the trends of green space in the three cities: specifically, the green spaces in Aarhus and Velika Gorica show an increasing trend, while the area of green space in

the Paris region decreases in all scenarios, with decreases of 5.3% (BAU), 5.9% (MLS), and 4.8% (EPS).

For the validation of land-cover simulation results, Table A2 shows the ROC values of the logistic regression results. The mean value of ROC for each land cover was greater than 0.8 across all six cities, indicating a good correlation and ability to explain land cover based on the selected driving factors. Table A3 shows the evaluation of the classification results for the six cities obtained by comparing the observed and simulated land cover in 2020 using 2000 random samples per city, where the mean value of the overall accuracy reached 0.8 and the mean value of the kappa value was 0.74 (Table A3). These figures indicate that the land-cover simulation model developed in this study can produce a convincing result.

### 4.2. Dynamics of GI and ES

#### 4.2.1. GI Fraction and Connectivity Changes

Figure 5 shows the changes in GIF and GIC distribution for the six cities from 2000 to 2020. At the city-wide scale, the GIF values of the three Chinese cities decrease significantly; the average of GIF in the three cities decreased from 85.5% in 2000 to 75.1% in 2020. The most significant reduction in GIF is in the new built-up area, where the average GIF of the three cities in the region decreases from 88% in 2000 to 57.8% in 2020. It is worth noting that the GIF and GIC values increase for most areas of the long-term built-up in Beijing and Shanghai between 2010 and 2020. It is also reflected in the regional averages; for example, from 2010 to 2020 in Shanghai, the GIF of the long-term built-up region increases from 25.6% to 28.1% and the GIC increases from 77.6 to 81. For the three European cities, the changes in GIF and GIC at the city scale are insignificant, with a slight decrease from 89% in 2000 to 86.4% in 2020 for the three cities. GIF, on the other hand, also shows a very small decrease from 97.1 in 2000 to 96.8 in 2020.

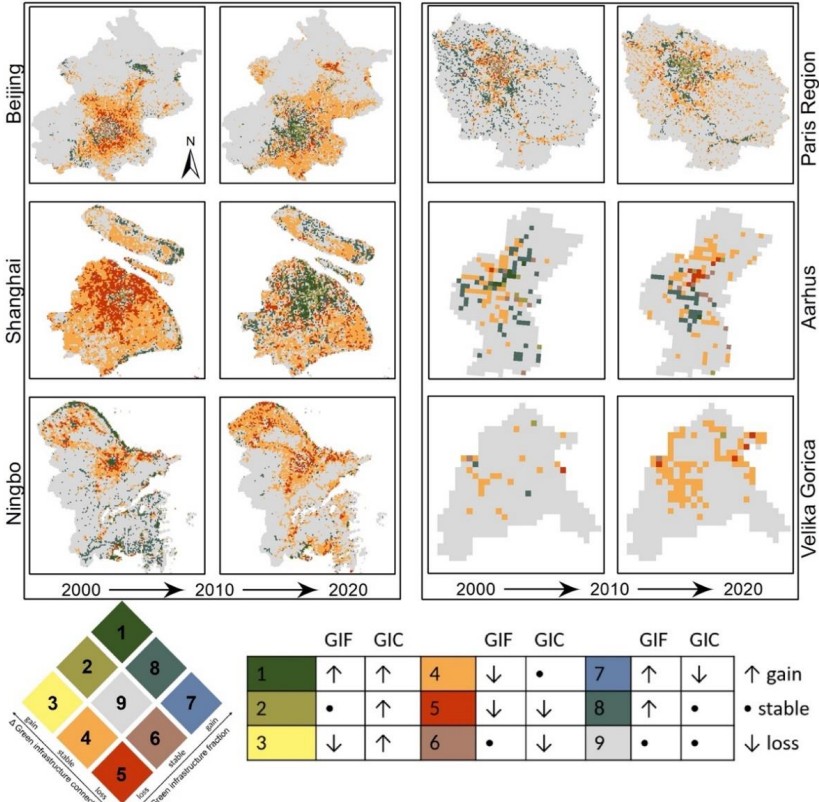

**Figure 5.** The spatial distribution of the combined changes in GIF and GIC for the two time periods (2000 to 2010 and 2010 to 2020), where indicators are considered to have increased or decreased significantly when the absolute value of the change is greater than 5%.

The GIF and GIC in different scenarios in 2030 show a big difference between cities in China and Europe (Figure 6). In China, the average GIFs for the three cities are 59.7% (MLS), 63.3% (BAU), and 66.8% (EPS). In the EPS scenario, GIF increased by 7.1% relative to the MLS scenario, and GIC increased by 18.7 relative to MLS. In the three European cities, the GIFs were 82.7% (MLS), 83% (BAU), and 83.4% (EPS), and the GICs were 90.3 (MLS), 91.3 (BAU), and 92.3 (EPS), respectively, with small differences across scenarios.

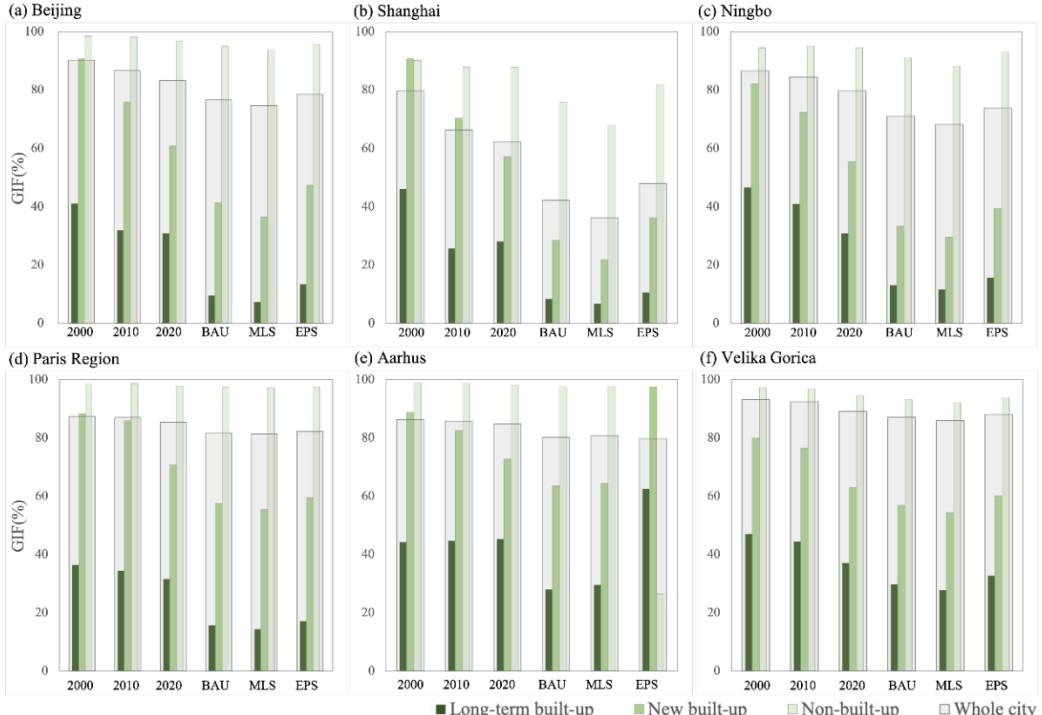

**Figure 6.** GIF in different urban development phases (long-term built-up, new built-up, and non-built-up) of each city from 2000 to 2030 and under the three BAU, MLS, and EPS scenarios for 2030.

4.2.2. Dynamics in Carbon Stock

Figures 7 and A4 show the distribution of carbon stock changes from 2000 to 2020 and 2020 to 2030 under different scenarios. Between 2000 and 2020, some areas of the long-term built-up areas in Beijing and Shanghai have greater carbon stocks, with increases in the southwestern and southeastern parts of Beijing and increases in Shanghai mainly in the outer ring green belt and Chongming Island. For Ningbo, the carbon stock shows a decreasing trend between 2000 and 2020. From 2020 to 2030 under various scenarios, there is a significant increase in carbon stock in the EPS scenario relative to the BAU and MLS. As for the three cities in Europe, from 2000 to 2020, carbon stock shows a stable trend in the Paris region, while in Aarhus, there is a strong increasing trend in carbon stock in the south of the city. In Velika Gorica, the carbon increase was significant throughout the region, especially in the forest area.

Figure A5 shows the changes of carbon stock in different urban development phases. Among the six cities, in 2020, Beijing has the largest carbon stock with 210.7 Mt, followed by the Paris region with 140 Mt. There is a slight increase in the long-term built-up carbon stock in Shanghai between 2010 and 2020, from 11.9 Mt to 12.4 Mt. Under different development scenarios, the most carbon stock is found in the EPS scenario. For example, in China, the total carbon stocks of the three cities in the EPS scenario are 7.1 Mt and 14.3 Mt more than those in the BAU and MLS scenarios, respectively. The Chinese cities generally show much more carbon in the newly built up areas compared with old built-up areas, when compared with European cities. Shanghai in particular shows a large proportion in newly built-up compared with the total study area (Figure A5).

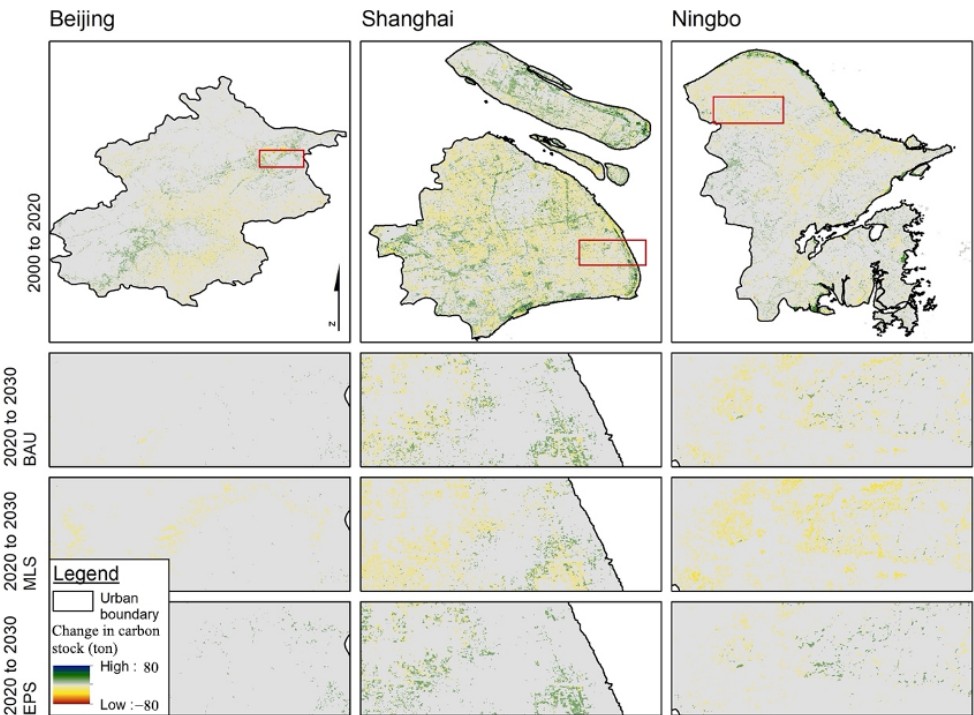

**Figure 7.** Carbon stock in Beijing, Shanghai, and Ningbo for 2000 to 2020 and 2020 to 2030 under three different scenarios (BAU, MLS, and EPS). Lower panels show detail within red boxes shown in upper panels.

4.2.3. Dynamics in $PM_{2.5}$ Removal

Figures 8 and A6 show the dynamics of $PM_{2.5}$ removal under different development scenarios from 2000 to 2020 and from 2020 to 2030. Between 2000 and 2020, both Shanghai in China and Velika Gorica in Europe exhibit large increases in $PM_{2.5}$ removal. The results for regions at different phases (Figure A7) of urban development show that Beijing has the highest $PM_{2.5}$ removal among the six cities. The level of $PM_{2.5}$ removal in Shanghai substantially increased from 2000 to 2020 and continued to increase slightly in 2030, especially under EPS. For Ningbo, the amount of $PM_{2.5}$ removal increased from 2000 to 2010 but then decreased in 2020. $PM_{2.5}$ removal in the Paris region experienced an increase up to 2010 and then a decrease in 2020, with insignificant differences in $PM_{2.5}$ removal among the three future scenarios. For Aarhus and Velika Gorica, their $PM_{2.5}$ removals are much lower compared with the other cities, with insignificant changes in the values of $PM_{2.5}$ removals for Aarhus in each year. Detailed information about the dynamics of GI and ES in different urban development stages and different years are available in the supporting materials (Tables A4–A7).

*4.3. Sensitivity of GI and ES to Land-Cover Changes*

Table 2 shows the SI of various indicators related to land-cover changes. Overall, most of the SIs for the four different indicators are less than 0, indicating that land-cover changes in the six cities have a negative impact on the coverage and connectivity of GI, as well as on the amount of carbon sequestration and air pollution removal. Based on observations from 2000 to 2020, land-cover change showed a positive effect on carbon stock and $PM_{2.5}$ removal from 2000 to 2010 in Beijing, while land cover in Shanghai continued to positively affect carbon stock and $PM_{2.5}$ removal from 2000 to 2020. In Ningbo, land-cover change had a positive effect on $PM_{2.5}$ removal only from 2000 to 2010. For the three European cities, land-cover change had a small positive effect on carbon stock and $PM_{2.5}$ removal between 2000 and 2020, while in Aarhus, the land cover had a positive effect on carbon stock and $PM_{2.5}$ removal between 2010 and 2020, where $SI_{carbon}$ was 0.02, and $SI_{pm2.5}$

was 4.16. In Velika Gorica, there was a continuous positive effect of land cover on both carbon stock and $PM_{2.5}$ removal between 2000 and 2020.

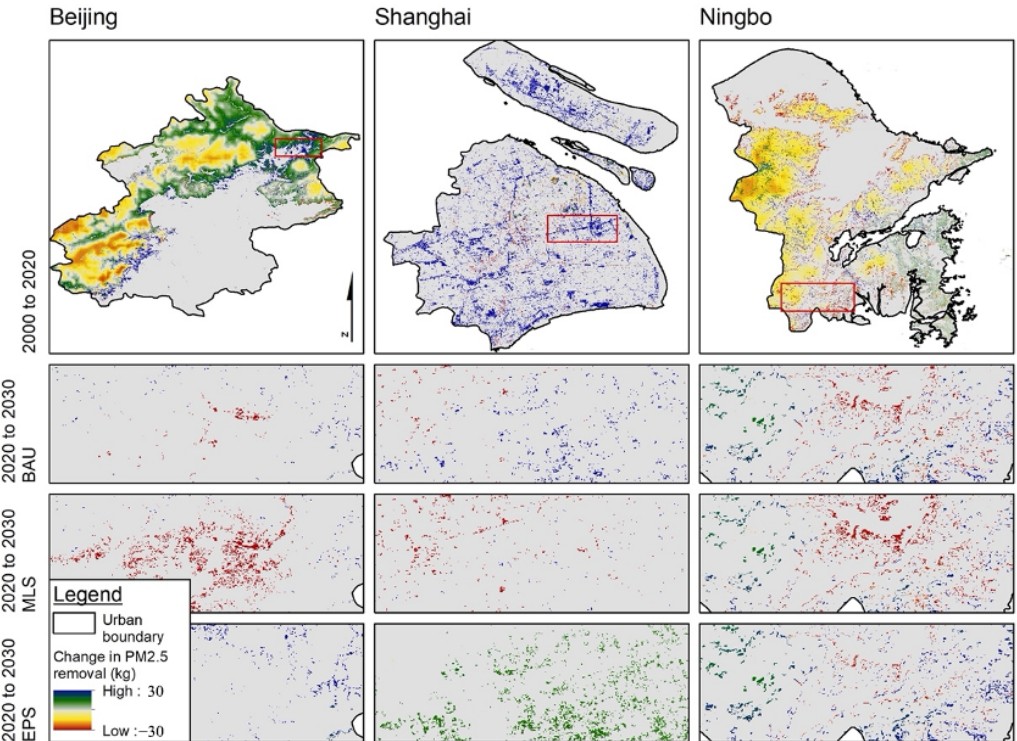

**Figure 8.** Distribution of $PM_{2.5}$ removal changes in Beijing, Shanghai, and Ningbo for 2000 to 2020 and 2020 to 2030 under three different scenarios. Lower panels show detail within red boxes shown in upper panels.

Under the three scenarios in 2030, land-cover change had a negative impact on all four indicators to varying degrees. The impact of land-cover change on ES was most pronounced under MLS, with an average SI of −1.95. It is worth noting that in the Paris region, land-cover change positively affects both carbon stock and $PM_{2.5}$ removal under EPS.

### 4.4. Equity of GI Distribution

In order to better understand changes in the distribution of GI in areas that rarely experience urban expansion, in Table 3, the GINI coefficients are shown as a function of the GIF distribution of long-term built-up areas on a 1 km × 1 km grid between 2000 and 2030. According to the results, GINI coefficients of GIF distributions from 2010 to 2020 decreased for Beijing and Shanghai, indicating a greater equity of GIF in the long-term built-up areas of the region. By contrast, for Ningbo, the GINI coefficients increased from 2000 to 2020, indicating an increasing inequity of GIF distributions. Among the three European cities, the GINI coefficients of the GIF distribution in the long-term built-up area vary less, and Velika Gorica has the smallest GINI coefficient of the GIF distribution in the stable built-up area, with a GINI value of 0.1 in 2020 and a GINI value of 0.24 in the EPS scenario in 2030. The GINI coefficient values increase the least in the EPS scenario in 2030, indicating that the EPS scenario is also a guarantee of the equity of the GIF distribution. In all cases, the GINI coefficients are greatest (i.e., greatest inequity) in the MLS scenario.

**Table 2.** Sensitivity index (SI) of the impact of land-cover change on GI and ESES changes in various stages. A higher SI value means that the indicator is more sensitive to land-cover change. When SI is greater than 0, it indicates that land-cover change has a positive effect on indicator change; otherwise, land-cover change has a negative effect on indicator change.

| City | SI | 2000–2010 | 2010–2020 | 2000–2020 | 2020–2030 (BAU) | 2020–2030 (MLS) | 2020–2030 (EPS) |
|---|---|---|---|---|---|---|---|
| Beijing | $SI_{GIF}$ | −0.30 | −0.27 | −0.50 | −1.56 | −1.40 | −1.30 |
| | $SI_{GIC}$ | −0.04 | −0.02 | −0.06 | −1.78 | −1.66 | −1.12 |
| | $SI_{Carbon}$ | −0.20 | 0.04 | −0.13 | −0.68 | −1.07 | −0.27 |
| | $SI_{PM2.5}$ | 5.06 | −2.32 | 0.54 | −1.91 | −3.83 | −2.66 |
| | $SI_{mean}$ | 1.13 | −0.64 | −0.04 | −1.48 | −1.99 | −1.34 |
| Shanghai | $SI_{GIF}$ | −0.63 | −0.22 | −0.52 | −1.88 | −2.00 | −1.69 |
| | $SI_{GIC}$ | −0.14 | 0.00 | −0.08 | −1.62 | −2.60 | −0.71 |
| | $SI_{Carbon}$ | −0.23 | 0.10 | −0.08 | −0.59 | −0.98 | −0.25 |
| | $SI_{PM2.5}$ | 7.75 | 6.72 | 19.22 | −3.11 | −4.19 | −4.06 |
| | $SI_{mean}$ | 1.69 | 1.65 | 4.63 | −1.80 | −2.44 | −1.68 |
| Ningbo | $SI_{GIF}$ | −0.12 | −0.31 | −0.38 | −1.37 | −1.35 | −1.52 |
| | $SI_{GIC}$ | 0.01 | −0.06 | −0.05 | −1.00 | −1.16 | −0.82 |
| | $SI_{Carbon}$ | −0.19 | 0.07 | −0.12 | −0.64 | −1.02 | −0.28 |
| | $SI_{PM2.5}$ | 1.12 | −1.80 | −0.88 | −1.27 | −2.16 | −1.24 |
| | $SI_{mean}$ | 0.21 | −0.53 | −0.36 | −1.07 | −1.42 | −0.96 |
| Paris Region | $SI_{GIF}$ | −0.04 | −0.22 | −0.23 | −1.03 | −1.05 | −1.01 |
| | $SI_{GIC}$ | −0.02 | 0.01 | −0.01 | −1.53 | −1.57 | −1.39 |
| | $SI_{Carbon}$ | 0.06 | −0.49 | −0.37 | −0.34 | −0.16 | 0.57 |
| | $SI_{PM2.5}$ | 1.10 | −4.06 | −2.85 | −0.51 | 0.93 | 2.36 |
| | $SI_{mean}$ | 0.27 | −1.19 | −0.86 | −0.85 | −0.46 | 0.13 |
| Aarhus | $SI_{GIF}$ | −0.03 | −0.17 | −0.15 | −3.22 | −2.75 | −3.85 |
| | $SI_{GIC}$ | 0.02 | −0.04 | −0.02 | −2.12 | −2.18 | −1.98 |
| | $SI_{Carbon}$ | −0.17 | 0.02 | −0.13 | −0.98 | −1.59 | −0.30 |
| | $SI_{PM2.5}$ | −2.04 | 4.16 | 0.53 | −1.64 | −9.69 | −13.17 |
| | $SI_{mean}$ | −0.55 | 0.99 | 0.06 | −1.99 | −4.05 | −4.82 |
| Velika Gorica | $SI_{GIF}$ | −0.06 | −0.18 | −0.19 | −0.27 | −0.39 | −0.14 |
| | $SI_{GIC}$ | 0.00 | −0.02 | −0.02 | −0.95 | −1.03 | −0.86 |
| | $SI_{Carbon}$ | 0.27 | 0.17 | 0.31 | −0.37 | −1.12 | −0.54 |
| | $SI_{PM2.5}$ | 0.07 | 1.07 | 0.94 | −0.44 | −2.73 | −2.34 |
| | $SI_{mean}$ | 0.07 | 0.26 | 0.26 | −0.51 | −1.31 | −0.97 |
| | **SI** | −10.00 | −6.00 | −2.00 | 2.00 | 6.00 | 10.00 |
| | | −15 | | | 0 | | 10 |

**Table 3.** GINI coefficients of GIF distribution for different scenarios from 2000 to 2030 for six urban long-term development areas.

| | 2000 | 2010 | 2020 | 2030 (BAU) | 2030 (MLS) | 2030 (EPS) |
|---|---|---|---|---|---|---|
| Beijing | 0.37 | 0.42 | 0.39 | 0.76 | 0.84 | 0.66 |
| Shanghai | 0.32 | 0.41 | 0.36 | 0.64 | 0.69 | 0.57 |
| Ningbo | 0.28 | 0.32 | 0.37 | 0.64 | 0.68 | 0.58 |
| Paris Region | 0.38 | 0.41 | 0.41 | 0.58 | 0.61 | 0.55 |
| Aarhus | 0.34 | 0.35 | 0.33 | 0.38 | 0.41 | 0.36 |
| Velika Gorica | 0.08 | 0.08 | 0.10 | 0.27 | 0.30 | 0.24 |

## 5. Discussion

Urban development stages and policies directly affect the quantity and distribution pattern of GI, which in turn affects the equity of distribution of ES. This study enables a robust multi-context prediction of future land cover in cities and provides an assessment of past and future GI and ES functions.

The selection of six contrasting cities in China and Europe exemplifies the evolution of ES for fairly typical sizes of towns, cities, and megacities to illustrate impacts of ongoing urbanization processes. On the one hand, due to the difference of urbanization process and stage between China and Europe, the dynamic changes in land cover and ES showed substantial differences for similar time points. Specifically, Europe is a highly urbanized continent, while China is progressing rapidly towards an urbanized country over the last decades. Therefore, urban expansion in the three Chinese cities caused significant damage

to the coverage and connectivity of urban GI and the ES it provides [53]. In contrast, national land cover and ES in Europe have shown more moderate changes.

Specifically, the multi-scenario analysis provides urban planners and policy makers with more dimensions of reference, which may guide urban planning policies by demonstrating the characteristics of ES distribution and their impacts under different scenarios. In addition to the extensive research on the spatial and temporal dynamics of ES, the resulting environmental equity is also receiving increasing attention because it is essential for equitable urban policy making [54]. Recent studies show that the loss of ES equity can affect not only ethnic groups of people differently [55], it is also essential in order to attain the social inclusion that is part of SDG 11 [56]. Quantifying ES distribution patterns helps increase the understanding of its contribution to environmental equity and use it as an important reference for reallocating ecological resources.

Moreover, the multi-regional analysis distinguishes between the impacts of urbanization on ES at different stages of urban development, i.e., long-term, new, and non-built-up areas [14,53], which is useful to increase the understanding of how ES are affected by different urbanization intensities and dynamics. The focus on long-term built-up areas also provides insight into the status and recovery of ES in urban centers that are not subject to urban sprawl or in the post-urbanization phase, and, as revealed in China, that there is potential to retrospectively improve GI provision at large scale within existing urban areas. For example, the values and distributional balance of GI and ES in long-term built-up areas in Beijing and Shanghai, China, improved between 2010 and 2020, indicating a return to green in these cities, which is a direct outcome of government policies [1].

In particular, in the early stages of urban development, such as from 2000 to 2010, most regions of China experienced rapid urbanization and urban sprawl, and many blue-green elements were directly replaced by built-up areas, resulting in a direct loss of GI coverage and ES. During the latter stages of development, GI and ES have improved in long-term built-up areas with a higher demand for ES, as seen in Beijing and Shanghai from 2010 to 2020. As compared to the three Chinese cities, the European cities have experienced longer-term development and slower urban expansion in the past decades, so the EPS scenario is more likely to maintain and optimize ES in Chinese cities in the future.

The study examined both GI and ES in order to explore the comprehensive effects of urbanization and policies on urban ecosystems. Specifically, a high GIF, for instance, contributes to an inclusive, resilient, and sustainable urban development [57]. In addition to regulating urban microclimate and reducing urban heat islands, adequate green space also prevents surface runoff and floods as well as provides residents with habitat, recreation, and cultural opportunities [58–60]. Furthermore, GIC reflects the loss of natural habitat mentioned in SDG 15.5 and its consequences for biodiversity [61,62]. This is because natural vegetation corridors provide adequate connectivity, allowing species to move freely and contributing to biodiversity preservation, while low connectivity isolates species and threatens biodiversity [63]. The benefit of carbon stock as one of the main ES is that the absorbed carbon dioxide from the air in urban vegetation is bound to organic carbon through photosynthesis and ultimately stored. As $PM_{2.5}$ does great harm to the health of urban residents, we analyzed this indicator as, to a certain extent, it is captured and removed through the atmospheric process of dry deposition to vegetation surfaces.

For the uncertainty of the study, because of the high heterogeneity of land cover in urban centers, there are inevitable errors in both the mapping and simulation of land cover and which further lead to uncertainty in ES assessment. These uncertainties in ES assessment have also been explored extensively in previous studies [64]. Different spatial resolutions and time lags between historical and future parameters, available for land-cover simulations, may lead to uncertainties in the results. In future studies, the accuracy of land-cover simulations can be effectively improved by using more high-resolution inputs and more homogeneously stored spatial information, such as new road plans and established future protected areas. Furthermore, interpreting results of the ecosystem service model outputs is not straightforward, since the models include other components. For example,

in the estimation of PM$_{2.5}$ removal, the model used in this study is sensitive to the initial concentration of PM$_{2.5}$ as well as the removal capacity of trees [65]. Therefore, interpreting the final results requires knowledge about how other aspects (e.g., air pollution levels) are changing at the same time. When estimating carbon stock, the carbon density parameters were obtained from the literature but could be improved with more detailed data collected for each city.

## 6. Conclusions

Urbanization processing and policies can directly affect urban land cover patterns and further influence ES. In this study, six cities of different sizes from China and Europe were selected as case areas, and a framework for an integrated assessment of urban ecosystem service dynamics under different development scenarios (BAU, MLS, and EPS) in the past and future was proposed. Additionally, this study focuses on the dynamics and changes in the variability and equity of GI and ES among different cities (Chinese and European cities) and within cities at different stages of development, as well as quantifying the sensitivity between changes in each indicator with respect to land-cover change. The main conclusions of the study are as follows: (1) The use of multi-source remote sensing data and the CLUE-S model can simulate future urban land cover distribution patterns under different development scenarios, and the simulation accuracy performs well in cities of different scales. (2) Urbanization levels in China and Europe are still at very different stages of development, not only in terms of the intensity of land-cover changes but also in terms of the characteristics of the changes in GI and ES. (3) Long-term built-up areas can be an important indicator of urban regeneration and ecosystem restoration; e.g., Beijing and Shanghai in China have seen significant improvements in both green space coverage and equity and ES in long-term built-up areas over the last decade. (4) In the future, the expansion of built-up areas will remain the main trend, and the loss of green space and arable land will be greatly reduced in EPS scenarios compared to BAU and MLS, while the green space cover in stable built-up areas is more fragile and should be a priority area for protection. The results obtained from this study can be used as an important reference for urban planners and policy makers at a later stage.

**Author Contributions:** Conceptualization, W.W. and E.B.; Data curation, X.L.; Formal analysis, W.W. and X.L.; Funding acquisition, E.B.; Investigation, W.W. and X.L.; Methodology, W.W., X.L., L.J. and E.B.; Project administration, E.B.; Software, W.W., X.L. and J.K.; Supervision, E.B.; Validation, W.W.; Visualization, W.W. and J.K.; Writing—original draft, W.W., X.L., J.K., L.J. and E.B.; Writing—review and editing, W.W., L.J. and E.B. All authors have read and agreed to the published version of the manuscript.

**Funding:** This research is supported jointly by the National Key Research and Development Project of China (Grant No. 2021YFE0193100), the European Union's Horizon 2020 research and innovation program (Grant No. 821016), the National Key Research and Development Program of China (Grant No. 2019YFA0607201), the Science and Technology Commission of Shanghai (Grant No. 19DZ1203405), and, finally, the China Scholarship Council (grant no. 202106100112).

**Conflicts of Interest:** The authors declare no conflict of interest.

## Appendix A

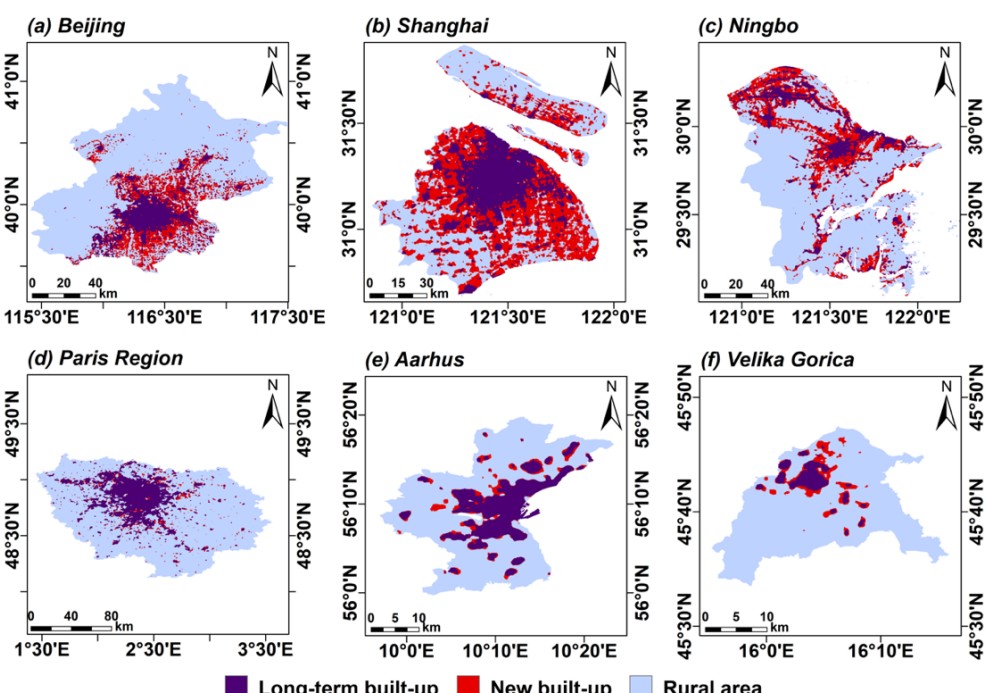

**Figure A1.** Urban-rural phases for all six cities in 2020.

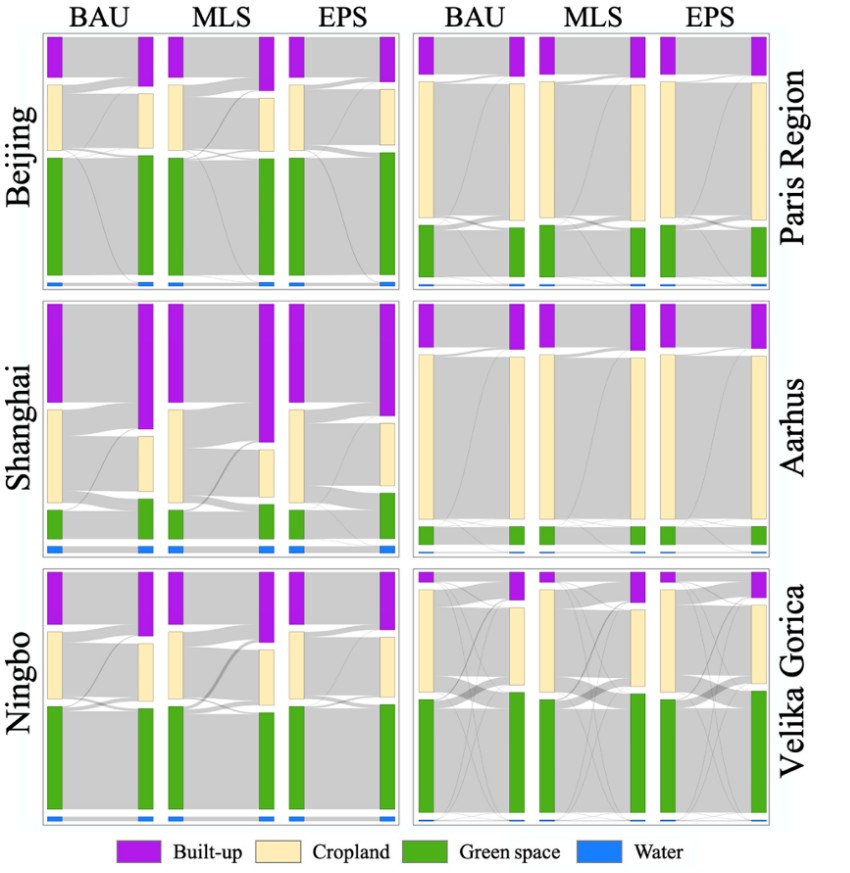

**Figure A2.** Land-cover transfer between 2020 and 2030 for various categories under different scenarios.

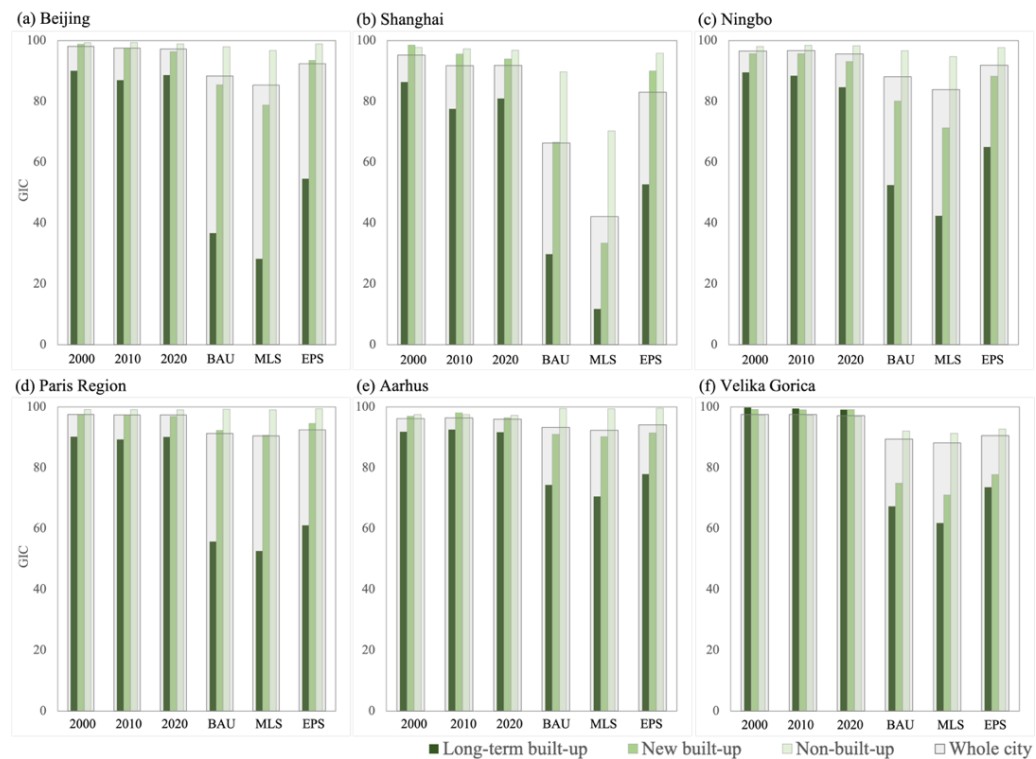

**Figure A3.** Statistics on the change of the average value of GIC in different regions of each city from 2000 to 2030, where BAU, MLS, and EPS represent three different scenarios for 2030.

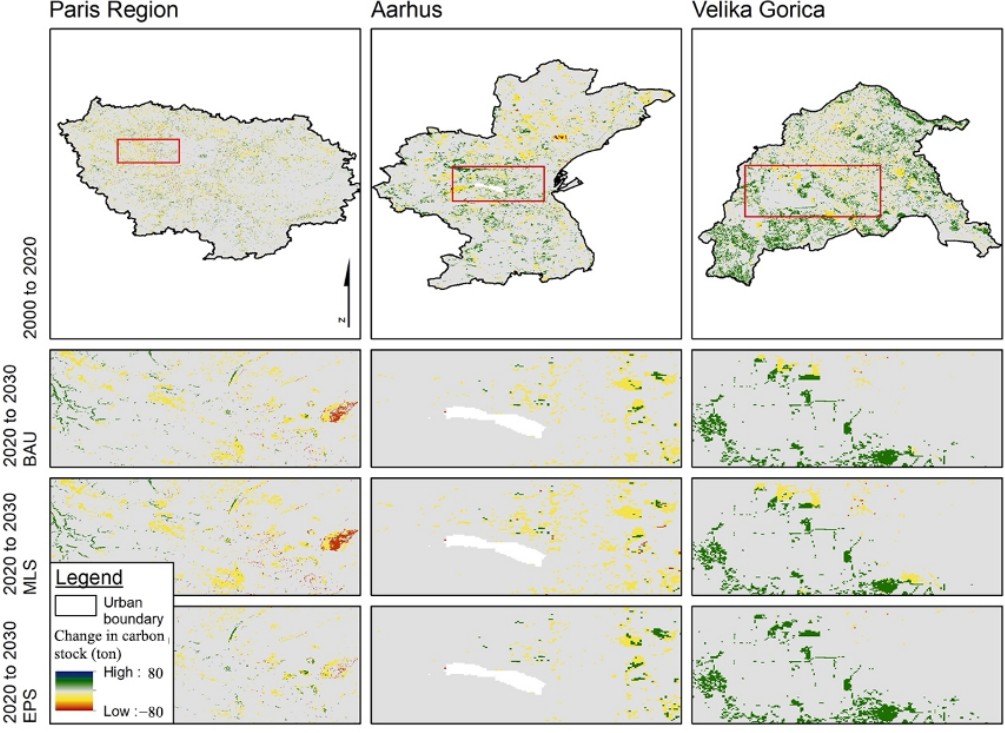

**Figure A4.** Changes in carbon stock in Paris region, Aarhus, and Velika Gorica for 2000 to 2020 and 2020 to 2030 under three different scenarios (BAU, MLS, and EPS).

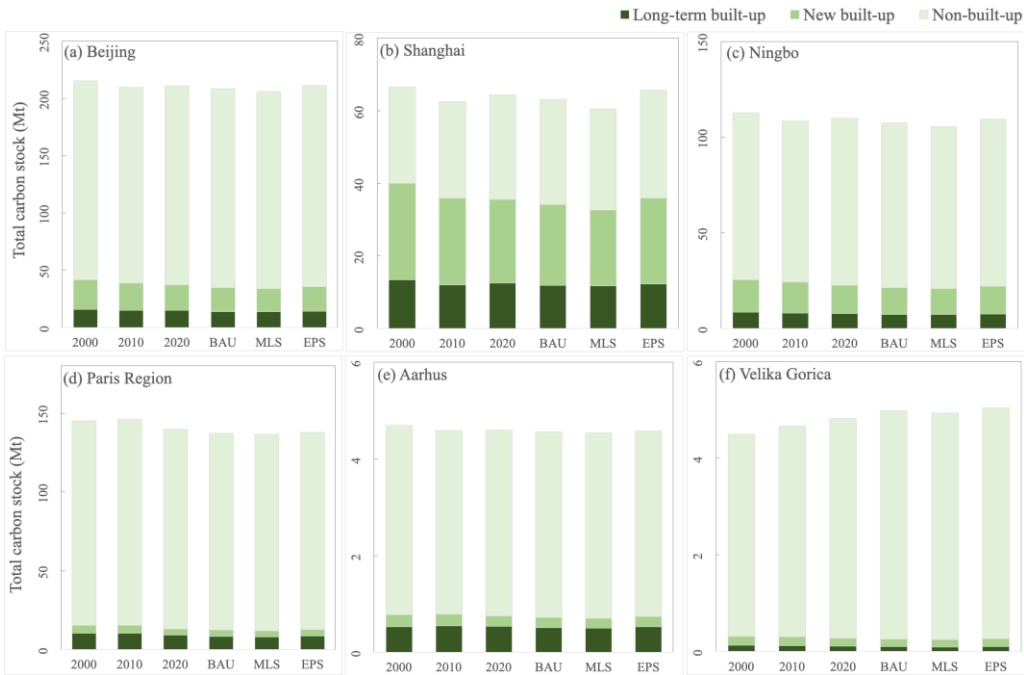

**Figure A5.** Total carbon stock statistics for different urban development phases of six cities from 2000 to 2030.

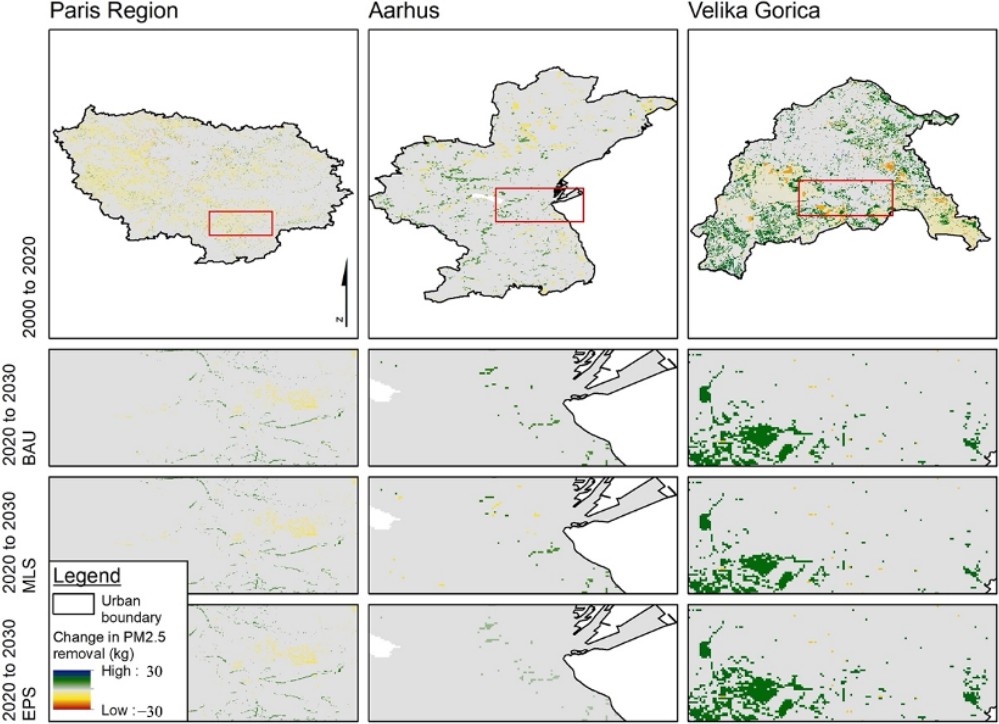

**Figure A6.** Changes in PM2.5 removal changes for 2000 to 2020 and 2020 to 2030 in Paris region, Aarhus, and Velika Gorica under three different scenarios (BAU, MLS, and EPS). Similar to carbon storage, PM2.5 removal is also greatly affected by land cover changes.

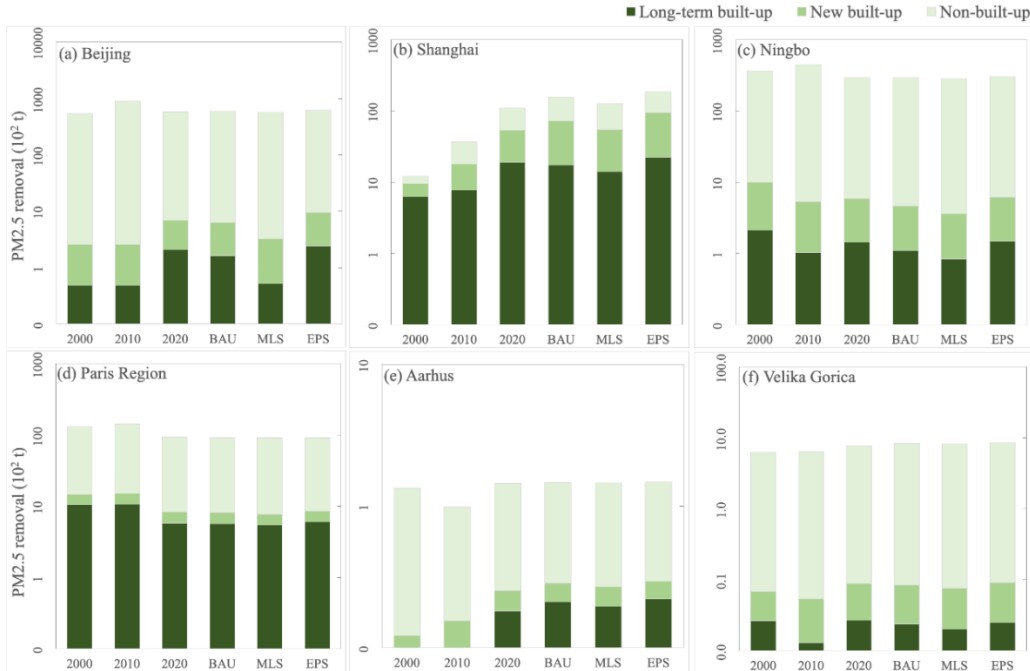

**Figure A7.** Total PM$_{2.5}$ removal for different development areas of six cities from 2000 to 2030.

**Table A1.** Carbon density of four pools in different land use types. (Unit: t/ha).

| Land Cover Types | C_above | | C_below | | C_soil | | C_dead | |
|---|---|---|---|---|---|---|---|---|
| | CN | EU | CN | EU | CN | EU | CN | EU |
| Built-up | 5 | 0 | 1 | 0 | 60 | 20.4 | 0 | 0 |
| Cropland | 8 | 4 | 1 | 0 | 97.84 | 102 | 0 | 0 |
| Green space | 40.54 | 56.4 | 20.27 | 12.97 | 92.96 | 130 | 12 | 16 |
| Water | 0 | 0 | 0 | 0 | 0 | 0 | 0 | 0 |

(C_above: aboveground carbon pool, C_below: belowground carbon pool, C_soil: soil organic carbon pool, C_dead: dead matters organic carbon pool.).

**Table A2.** ROC values for simulated land cover.

| City | Built-Up | Cropland | Forest | Water |
|---|---|---|---|---|
| Beijing | 0.94 | 0.89 | 0.98 | 0.88 |
| Shanghai | 0.90 | 0.86 | 0.72 | 0.77 |
| Ningbo | 0.91 | 0.84 | 0.96 | 0.86 |
| Paris Region | 0.96 | 0.81 | 0.82 | 0.95 |
| Aarhus | 0.91 | 0.79 | 0.90 | 0.97 |
| Velika Gorica | 0.89 | 0.82 | 0.92 | 0.99 |
| Mean value | 0.92 | 0.84 | 0.88 | 0.90 |

**Table A3.** Accuracy of land-cover observation in 2020 compared with simulation results.

| City | Beijing | Shanghai | Ningbo | Paris Region | Velika Gorica | Aarhus | Mean Value |
|---|---|---|---|---|---|---|---|
| OA | 0.77 | 0.62 | 0.88 | 0.86 | 0.86 | 0.82 | 0.80 |
| Kappa | 0.70 | 0.49 | 0.86 | 0.81 | 0.81 | 0.76 | 0.74 |

**Table A4.** Statistics of the GIF in different areas of each city from 2000 to 2030.

| City | Areas | 2000 | 2010 | 2020 | BAU | MLS | EPS |
|------|-------|------|------|------|-----|-----|-----|
| Beijing | LB | 41.05 | 31.90 | 30.78 | 9.49 | 7.35 | 13.47 |
| | NewB | 90.89 | 75.98 | 60.84 | 41.44 | 36.44 | 47.46 |
| | NonB | 98.54 | 98.26 | 96.70 | 95.05 | 93.79 | 95.79 |
| | Whole city | 90.32 | 86.80 | 83.30 | 76.62 | 74.70 | 78.54 |
| Shanghai | LB | 46.12 | 25.60 | 28.10 | 8.33 | 6.78 | 10.56 |
| | NewB | 90.67 | 70.29 | 57.12 | 28.47 | 21.94 | 36.29 |
| | NonB | 90.16 | 87.83 | 87.88 | 75.87 | 67.95 | 81.98 |
| | Whole city | 79.73 | 66.38 | 62.16 | 42.15 | 36.28 | 47.96 |
| Ningbo | LB | 46.58 | 40.97 | 30.80 | 13.13 | 11.62 | 15.69 |
| | NewB | 82.32 | 72.43 | 55.51 | 33.31 | 29.67 | 39.44 |
| | NonB | 94.49 | 95.18 | 94.53 | 91.16 | 88.33 | 93.14 |
| | Whole city | 86.57 | 84.52 | 79.68 | 71.07 | 68.23 | 73.90 |
| Paris Region | LB | 36.49 | 34.45 | 31.63 | 15.70 | 14.43 | 17.19 |
| | NewB | 88.28 | 85.88 | 70.84 | 57.60 | 55.44 | 59.47 |
| | NonB | 98.50 | 98.65 | 97.86 | 97.41 | 97.21 | 97.57 |
| | Whole city | 87.32 | 87.01 | 85.38 | 81.79 | 81.33 | 82.24 |
| Aarhus | LB | 44.30 | 44.63 | 45.23 | 27.97 | 29.58 | 62.55 |
| | NewB | 88.75 | 82.54 | 72.70 | 63.57 | 64.31 | 97.52 |
| | NonB | 98.99 | 98.73 | 97.89 | 97.61 | 97.64 | 26.47 |
| | Whole city | 86.20 | 85.76 | 84.77 | 80.22 | 80.64 | 79.77 |
| Velika Gorica | LB | 47.09 | 44.48 | 37.05 | 29.86 | 27.73 | 32.79 |
| | NewB | 79.99 | 76.47 | 62.92 | 56.82 | 54.49 | 60.15 |
| | NonB | 97.24 | 96.78 | 94.44 | 93.05 | 92.13 | 93.83 |
| | Whole city | 93.14 | 92.37 | 89.07 | 87.05 | 85.98 | 88.11 |
| China | LB | 44.58 | 32.82 | 29.90 | 10.32 | 8.58 | 13.24 |
| | NewB | 87.96 | 72.90 | 57.83 | 34.41 | 29.35 | 41.07 |
| | NonB | 94.40 | 93.76 | 93.04 | 87.36 | 83.35 | 90.30 |
| | Whole city | 85.54 | 79.23 | 75.05 | 63.28 | 59.74 | 66.80 |
| Europe | LB | 42.63 | 41.19 | 37.97 | 24.51 | 23.91 | 37.51 |
| | NewB | 85.67 | 81.63 | 68.82 | 59.33 | 58.08 | 72.38 |
| | NonB | 98.24 | 98.05 | 96.73 | 96.02 | 95.66 | 72.63 |
| | Whole city | 88.89 | 88.38 | 86.40 | 83.02 | 82.65 | 83.37 |

(Units: %) Where BAU, MLS, and EPS represent three different scenarios for 2030. LB, NewB, and NonB represent different developed phrases of city, named long-term built-up, new built-up, and non-built-up.

**Table A5.** Statistics of the GIC in different areas of each city from 2000 to 2030.

| City | Areas | 2000 | 2010 | 2020 | BAU | MLS | EPS |
|------|-------|------|------|------|-----|-----|-----|
| Beijing | LB | 90.03 | 86.94 | 88.65 | 36.65 | 28.19 | 54.51 |
| | NewB | 98.77 | 97.43 | 96.38 | 85.46 | 78.80 | 93.46 |
| | NonB | 99.34 | 99.39 | 98.86 | 97.99 | 96.74 | 98.88 |
| | Whole city | 98.06 | 97.51 | 97.19 | 88.29 | 85.32 | 92.41 |
| Shanghai | LB | 86.31 | 77.55 | 80.96 | 29.74 | 11.78 | 52.70 |
| | NewB | 98.51 | 95.52 | 93.89 | 66.58 | 33.41 | 89.98 |
| | NonB | 97.76 | 97.20 | 96.80 | 89.71 | 70.24 | 95.82 |
| | Whole city | 95.21 | 91.73 | 91.81 | 66.30 | 42.11 | 83.00 |
| Ningbo | LB | 89.56 | 88.43 | 84.65 | 52.45 | 42.40 | 64.96 |
| | NewB | 95.67 | 95.69 | 93.05 | 80.08 | 71.29 | 88.25 |
| | NonB | 98.05 | 98.44 | 98.24 | 96.62 | 94.74 | 97.65 |
| | Whole city | 96.56 | 96.69 | 95.58 | 88.05 | 83.84 | 91.85 |

**Table A5.** *Cont.*

| City | Areas | 2000 | 2010 | 2020 | BAU | MLS | EPS |
|------|-------|------|------|------|-----|-----|-----|
| Paris Region | LB | 90.18 | 89.33 | 90.12 | 55.75 | 52.62 | 61.03 |
| | NewB | 97.55 | 97.35 | 96.88 | 92.29 | 90.65 | 94.55 |
| | NonB | 99.13 | 99.12 | 99.03 | 99.23 | 98.98 | 99.41 |
| | Whole city | 97.48 | 97.32 | 97.37 | 91.26 | 90.45 | 92.42 |
| Aarhus | LB | 91.86 | 92.59 | 91.71 | 74.34 | 70.57 | 77.86 |
| | NewB | 96.95 | 98.03 | 96.44 | 91.02 | 90.23 | 91.48 |
| | NonB | 97.45 | 97.47 | 97.21 | 99.47 | 99.42 | 99.50 |
| | Whole city | 96.14 | 96.38 | 95.91 | 93.25 | 92.30 | 94.10 |
| Velika Gorica | LB | 99.73 | 99.48 | 99.06 | 67.32 | 61.82 | 73.61 |
| | NewB | 99.17 | 99.02 | 99.09 | 74.89 | 71.07 | 77.77 |
| | NonB | 97.25 | 97.23 | 96.86 | 92.04 | 91.32 | 92.71 |
| | Whole city | 97.53 | 97.48 | 97.14 | 89.39 | 88.18 | 90.56 |
| China | LB | 88.63 | 84.31 | 84.75 | 39.61 | 27.46 | 57.39 |
| | NewB | 97.65 | 96.21 | 94.44 | 77.37 | 61.17 | 90.56 |
| | NonB | 98.38 | 98.34 | 97.97 | 94.77 | 87.24 | 97.45 |
| | Whole city | 96.61 | 95.31 | 94.86 | 80.88 | 70.42 | 89.09 |
| Europe | LB | 93.92 | 93.80 | 93.63 | 65.80 | 61.67 | 70.83 |
| | NewB | 97.89 | 98.13 | 97.47 | 86.07 | 83.98 | 87.93 |
| | NonB | 97.94 | 97.94 | 97.70 | 96.91 | 96.57 | 97.21 |
| | Whole city | 97.05 | 97.06 | 96.81 | 91.30 | 90.31 | 92.36 |

**Table A6.** Total carbon storage statistics for different areas of each city from 2000 to 2030. (Unit: Mt).

| City | Areas | 2000 | 2010 | 2020 | BAU | MLS | EPS |
|------|-------|------|------|------|-----|-----|-----|
| Beijing | LB | 15.88 | 15.07 | 14.95 | 13.89 | 13.70 | 14.22 |
| | NewB | 25.62 | 23.73 | 21.98 | 20.75 | 20.21 | 21.43 |
| | NonB | 173.69 | 170.64 | 173.76 | 173.82 | 172.03 | 175.32 |
| | Whole city | 215.18 | 209.44 | 210.69 | 208.45 | 205.94 | 210.96 |
| Shanghai | LB | 13.38 | 11.92 | 12.41 | 11.88 | 11.64 | 12.22 |
| | NewB | 26.60 | 23.99 | 23.19 | 22.22 | 20.96 | 23.72 |
| | NonB | 26.68 | 26.70 | 28.88 | 29.09 | 27.92 | 29.84 |
| | Whole city | 66.70 | 62.60 | 64.50 | 63.20 | 60.50 | 65.80 |
| Ningbo | LB | 8.35 | 7.90 | 7.58 | 7.28 | 7.20 | 7.41 |
| | NewB | 16.93 | 16.16 | 14.87 | 13.92 | 13.61 | 14.43 |
| | NonB | 87.42 | 84.35 | 87.40 | 86.21 | 84.63 | 87.54 |
| | Whole city | 112.70 | 108.40 | 109.84 | 107.41 | 105.44 | 109.38 |
| Paris Region | LB | 10.26 | 10.40 | 8.99 | 8.31 | 8.01 | 8.63 |
| | NewB | 5.10 | 4.98 | 4.05 | 3.88 | 3.78 | 3.97 |
| | NonB | 129.95 | 130.66 | 126.94 | 125.16 | 124.93 | 125.36 |
| | Whole city | 145.30 | 146.05 | 139.98 | 137.34 | 136.72 | 137.97 |
| Aarhus | LB | 0.53 | 0.54 | 0.54 | 0.51 | 0.49 | 0.53 |
| | NewB | 0.25 | 0.24 | 0.21 | 0.21 | 0.21 | 0.21 |
| | NonB | 3.91 | 3.81 | 3.85 | 3.84 | 3.84 | 3.84 |
| | Whole city | 4.69 | 4.59 | 4.60 | 4.56 | 4.54 | 4.58 |
| Velika Gorica | LB | 0.13 | 0.12 | 0.11 | 0.10 | 0.09 | 0.10 |
| | NewB | 0.19 | 0.18 | 0.16 | 0.15 | 0.15 | 0.16 |
| | NonB | 4.18 | 4.36 | 4.55 | 4.74 | 4.69 | 4.78 |
| | Whole city | 4.49 | 4.67 | 4.82 | 4.99 | 4.93 | 5.04 |
| China | LB | 37.61 | 34.89 | 34.94 | 33.05 | 32.54 | 33.85 |
| | NewB | 69.15 | 63.88 | 60.04 | 56.89 | 54.78 | 59.58 |
| | NonB | 287.79 | 281.69 | 290.04 | 289.12 | 284.58 | 292.70 |
| | Whole city | 394.58 | 380.44 | 385.03 | 379.06 | 371.88 | 386.14 |

**Table A6.** *Cont.*

| City | Areas | 2000 | 2010 | 2020 | BAU | MLS | EPS |
|------|-------|------|------|------|-----|-----|-----|
| Europe | LB | 10.92 | 11.06 | 9.64 | 8.92 | 8.59 | 9.26 |
| | NewB | 5.54 | 5.40 | 4.42 | 4.24 | 4.14 | 4.34 |
| | NonB | 138.04 | 138.83 | 135.34 | 133.74 | 133.46 | 133.98 |
| | Whole city | 154.48 | 155.31 | 149.40 | 146.89 | 146.19 | 147.59 |

**Table A7.** Total carbon storage statistics for different areas of each city from 2000 to 2030. (Unit: $10^2$ t).

| City | Areas | 2000 | 2010 | 2020 | BAU | MLS | EPS |
|------|-------|------|------|------|-----|-----|-----|
| Beijing | LB | 486.47 | 488.41 | 2110.37 | 1626.17 | 522.04 | 2431.99 |
| | NewB | 2111.61 | 2104.90 | 4852.52 | 4689.54 | 2743.50 | 7037.47 |
| | NonB | 537,607.45 | 894,294.79 | 578,345.61 | 585,690.19 | 564,301.11 | 605,787.67 |
| | Whole city | 540,205.52 | 896,888.10 | 585,308.49 | 592,005.90 | 567,566.65 | 615,257.13 |
| Shanghai | LB | 6.21 | 7.75 | 18.97 | 17.42 | 14.01 | 22.40 |
| | NewB | 3.23 | 10.15 | 34.41 | 54.91 | 39.94 | 72.48 |
| | NonB | 2.57 | 18.88 | 56.41 | 85.24 | 73.15 | 90.61 |
| | Whole city | 12.02 | 36.79 | 109.80 | 157.58 | 127.09 | 185.49 |
| Ningbo | LB | 2.11 | 1.03 | 1.44 | 1.09 | 0.84 | 1.48 |
| | NewB | 7.79 | 4.23 | 4.47 | 3.52 | 2.76 | 4.63 |
| | NonB | 356.23 | 443.20 | 291.98 | 287.76 | 278.42 | 296.69 |
| | Whole city | 366.13 | 448.45 | 297.89 | 292.37 | 282.02 | 302.79 |
| Paris Region | LB | 10.48 | 10.63 | 5.75 | 5.72 | 5.41 | 6.03 |
| | NewB | 4.14 | 4.33 | 2.49 | 2.37 | 2.31 | 2.44 |
| | NonB | 115.24 | 127.19 | 85.05 | 82.98 | 82.74 | 83.19 |
| | Whole city | 129.86 | 142.15 | 93.28 | 91.07 | 90.46 | 91.66 |
| Aarhus | LB | 0.08 | 0.10 | 0.18 | 0.21 | 0.20 | 0.22 |
| | NewB | 0.04 | 0.06 | 0.07 | 0.07 | 0.07 | 0.07 |
| | NonB | 1.22 | 0.83 | 1.20 | 1.19 | 1.19 | 1.18 |
| | Whole city | 1.35 | 0.99 | 1.45 | 1.47 | 1.46 | 1.48 |
| Velika Gorica | LB | 0.03 | 0.01 | 0.03 | 0.02 | 0.02 | 0.02 |
| | NewB | 0.04 | 0.04 | 0.06 | 0.06 | 0.06 | 0.07 |
| | NonB | 6.24 | 6.32 | 7.62 | 8.28 | 8.19 | 8.36 |
| | Whole city | 6.31 | 6.37 | 7.71 | 8.36 | 8.27 | 8.45 |
| China | LB | 494.79 | 497.19 | 2130.78 | 1644.68 | 536.89 | 2455.87 |
| | NewB | 2122.63 | 2119.28 | 4891.40 | 4747.97 | 2786.20 | 7114.58 |
| | NonB | 537,966.25 | 894,756.87 | 578,694.00 | 586,063.19 | 564,652.68 | 606,174.97 |
| | Whole city | 540,583.67 | 897,373.34 | 585,716.18 | 592,455.85 | 567,975.76 | 615,745.41 |
| Europe | LB | 10.59 | 10.74 | 5.96 | 5.95 | 5.63 | 6.27 |
| | NewB | 4.22 | 4.43 | 2.62 | 2.50 | 2.44 | 2.58 |
| | NonB | 122.70 | 134.34 | 93.87 | 92.45 | 92.12 | 92.73 |
| | Whole city | 137.52 | 149.51 | 102.44 | 100.90 | 100.19 | 101.59 |

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
