# Peer review of "A European-Chinese Exploration: Part 2—Urban Ecosystem Service Patterns, Processes, and Contributions to Environmental Equity under Different Scenarios"

_remotesensing, doi:10.3390/rs14143488_

Round 1

Reviewer 1 Report

Article review:

(1) Title suggestion:

A European-Chinese exploration: Part 2 - Comparative assessment of dynamics in urban green infrastructure and ecosystem services under different scenarios

(2) Introduction

Summarize the objectives. Become the text clear (pg 3):

- “Therefore, in this paper, we use ES models to assess the impact of different urbanization patterns on selected ES”: It is an objective.

- “We then use the CLUE-S model to simulate future urbanization patterns on both continents under a range of policy scenarios and to assess the impact of future urban distribution patterns on ES”: It is also an objective.

- “The main objectives of this research are as follow - (1) Predict ES distribution patterns over the next decade under three various scenarios. (2) Analyze the differences and characteristics of the relative ES dynamics provided by different phases of development. (3) Exploring changes in the equity of ES distribution during urbanization. (4) Exploring how land-cover changes have led to differences in green infrastructure (GI) and ES changes”: It is also four other objectives.

(3) Dataset and materials

Figure 1 - Become the figure clear (pg 4). “Our analysis comprises three Chinese cities (Beijing, Shanghai and Ningbo) as well as three European Cities (Paris Region - France, Aarhus - Denmark and Velika Gorica - Croatia)”. In the central figure, only the land use maps, and in a small figure, the cities localization in China, France, Denmark, and Croatia.

(4) Results

- ”Figure 3. Land-cover observation results for 2020 and simulated land cover maps for 2030 under three scenarios for three cities in China”. In the image scale, the changes are visible only in Shanghai.

- “Figure 4. Land-cover observation results for 2020 and simulated land cover maps for 2030 under three scenarios for three cities in Europe”. In the image scale, there aren´t changes visible.

- Suggestion: The text presented in “(4.1) In the image scale, there aren´t changes visible” could be adjusted, and the numerical information arranged in a table.

- (pg. 11) “Table A.2 shows the ROC values ..... Table A.3 shows the evaluation of the classification”. This information is on page 28.

- (pg. 14) “Figure 7. Carbon stock in Beijing, Shanghai and Ningbo for 2000 to 2020 and 2020 to 2030 under three different scenarios (BAU, MLS and EPS). Lower panels show detail within red boxes shown in upper panels”. In the image scale, the information is confusing.

(5) Discussion

- Suggestion: The discussion could follow the objectives order - ecosystem service distribution patterns in the various scenarios; the relative ecosystem service dynamics in the different phases of development; the equity of ecosystem service distribution during urbanization; and the differences in green infrastructure and ecosystem service changes. And the most relevant contrast and similarities between Chinese cities (Beijing, Shanghai, and Ningbo) and European Cities (Paris Region - France, Aarhus - Denmark, and Velika Gorica - Croatia).

Author Response

We sincerely appreciate the reviewer’s valuable comments and helpful suggestions on this manuscript. We have responded to all the comments point-by-point and made corresponding changes in the manuscript as highlighted in red color. The detailed responses to all the comments are as follows (the reviewer’s comments are in black and our replies are in blue, and the number of lines in the answer corresponds to the number of lines in the file "Manuscript with revision marks"). The changed text is marked in purple.

Reviewer 2 Report

Overall, the authors have employed different simulation and prediction models to reveal the urban landscape scenarios under different policies. The manuscript is well-organized, and I enjoy the figures the authors created since they were very good at picking the good color palettes. I didn't see significant flaws in the framework of this research, but there are some problems the authors should pay attention to and some questions to be answered.

1. I don't like your current title. Why did you put "Part 2" in the title? Also, for the workload of your research, you should create a more informative and attractive title.

2. In the abstract, the authors claimed, "However, it is unclear whether and to what extent .... and government policies". I see this statement as a little bit arbitrary. I don't think your current literature review introduced in the "Introduction" can support this.

3. There are some extra spaces between sentences. For example, in the third line in the first paragraph, before "ES provided". Such a tiny error should be corrected to improve the readability.

4. In the 2.1. Study area section, you can list the specific population of Paris.

5. This question is about the methodology for studying city selection. So, for China, you chose Beijing, Shanghai, and Ningbo; the first two are megacities in China. For Europe, you choose a megacity, a typical mid-sized city, and a town. Do you think these six cities are comparable? Megacity - megacity, megacity - mid-sized city, and mid-sized city - town? 

6. 2.2.2. OpenStreetMap, not "Open Street Map".

7. In Table 1, why did you exclude GDP for the socio-economic factors?

8. 2.2.3. three Chinese cities, not "Chines", typo.

9. In the third paragraph of 3.1.1., you said "For the location characteristics, xxx, including terrain, climate, ... and policy related factors (Table 1)". Yet, I didn't find any climatic factors in Table 1, neither did the policy related factors.

10. An extra space before "Finally, the land cover xxx".

11. The format of your Eq 1(a) and (b) needed to be adjusted, also please explain the symbols in the equations even though the equations are widely-adopted.

12. For GI fraction, why did you perform the computation at 1-km resolution instead of 60-m?

13. In 3.2.3, what are "urban living labs"?

14. For Equation 5, how did you determine the coefficients as "1.1664", and "0.4837"? If you use these numbers from others' work, please cite.

Author Response

(The authors gave the same response as above.)

Reviewer 3 Report

The manuscript is focus on assessment of dynamics in urban green infrastructure and ecosystem services under different scenarios using various models, as well as quantifying the sensitivity between changes in each indicator with respect to land cover change. The results consist of land cover simulation result, assessment of dynamics of green infrastructure and ecosystem services, assessment of green infrastructure and ecosystems services to land-cover changes, and assessment of equity of green infrastructure distribution. The use of multi-source remote sensing data and models can stimulate future urban land cover distribution patterns under different development scenarios. The results can be used as an important references for urban planners and policy makers. The aim of the study, methods and results are clearly described and the manuscript has high scientific soundness, so I belive it find interest to many readers. I have only small recommendation to cited references, if possible - try to avoid citation of web-page in the manuscript, cite web-page in accordance with the guidelines of the journal instructions to authors.

Author Response

(The authors gave the same response as above.)

Reviewer 4 Report

The paper aimed to show the time series of ES and GI in Chinese and European cities in a comparative manner.

The analyses are well done, but it is not clear to me what is the significant contribution of the study, especially to a policy aspect. The results are almost predictable in a qualitative sense, considering the two regions' circumstances. What is the point of showing the quantitative and geographical assessments of the situations for policy formation? Do the Chinese cities consider the European cities' indexed values as a goal? Further discussion is expected with these rich assessment outcomes.

Author Response

(The authors gave the same response as above.)

Reviewer 5 Report

Urban areas are human-environment systems that depend fundamentally on ecosystems, and thus require an understanding of the management of urban ecosystem services to ensure sustainable urban planning. This manuscript contain lots of work and definitely a good contribution to understand the assessment of dynamics in urban green infrastructure and ecosystem services under different scenarios. However, I could not understand well what was the actual purpose to compare two different climate setting region to fulfill the objectives.  I also made some suggestion , which may help to improve this manuscript. 

Since author(s) used many techniques/methods in this manuscript, I would suggest to keep simple and provide step by step link of techniques/methods, which is partially done by author(s). I would  like to suggest please explain in detail about the method/techniques rather than the importance of methods/techniques. How author(s) link all these mentioned methods to assessment of dynamics in urban green infrastructure and ecosystem services under different scenarios. It seems the author(s) just analysed different issues in urban ecosystem separately.

"landscape index algorithm" used once by author(s) in the manuscript. 

The link does not work of The World Bank.

Why author(s) used green infrastructure (GI) fraction and connectivity indicators to justify the urban ecosystem services? is there another techniques as a proxy ( indicator)  too? 

Please have a look below link of the manuscript: connectivity and Multicollinearity between selected factors

https://doi.org/10.3390/rs13204090

https://doi.org/10.3390/w14030402

https://iopscience.iop.org/article/10.1088/1742-6596/949/1/012009

Please use urban ES through out in the manuscript. for example author used development of ES is incorrect in this manuscript, I guess. 

As author(s) mentioned in the manuscript that policy decisions can also influence management and decision-making to change the distribution patterns of ES. what was the policy factor in table 1? need to justify. 

How author come up using different resolution of dataset in table 1. is it okay using different resolution for assessment purposes? 

What was the base to choose/select the variables ? what about other factors (external factors)?

Methodology section can be improved more. It is better to explain step by step about data sets and associated techniques, partially done by author(s). 

Please add the implication/limitation of such studies using different resolution dataset and briefly indicate the advantage or disadvantage ? 

Please make sure all dash, space, a hyphen, en dash, and capital words would be appropriate throughout the manuscript.

Please make sure the font size is in the table and figure. 

Please make sure the abbreviation is clearly defined in the text.

Please avoid long and unnecessary sentences through out in the manuscript. 

Author Response

(The authors gave the same response as above.)

Round 2

Reviewer 5 Report

I think manuscript can be published after editor decision. 

Please adjust the references style according to journal guideline.